# A critical role for HNF4α in polymicrobial sepsis-associated metabolic reprogramming and death

Céline Van Dender [ID][1,2], Steven Timmermans[1,2], Ville Paakinaho [ID][3], Tineke Vanderhaeghen [ID][1,2], Jolien Vandewalle[1,2], Maarten Claes[4], Bruno Garcia[5,6], Bart Roman[4], Jan De Waele [ID][7,8], Siska Croubels [ID][9], Karolien De Bosscher[10,11], Philip Meuleman[12], Antoine Herpain [ID][5,13], Jorma J Palvimo [ID][3] & Claude Libert [ID][1,2 ✉]

## Abstract

In sepsis, limited food intake and increased energy expenditure induce a starvation response, which is compromised by a quick decline in the expression of hepatic PPARα, a transcription factor essential in intracellular catabolism of free fatty acids. The mechanism upstream of this PPARα downregulation is unknown. We found that sepsis causes a progressive hepatic loss-of-function of HNF4α, which has a strong impact on the expression of several important nuclear receptors, including PPARα. HNF4α depletion in hepatocytes dramatically increases sepsis lethality, steatosis, and organ damage and prevents an adequate response to IL6, which is critical for liver regeneration and survival. An HNF4α agonist protects against sepsis at all levels, irrespectively of bacterial loads, suggesting HNF4α is crucial in tolerance to sepsis. In conclusion, hepatic HNF4α activity is decreased during sepsis, causing PPARα downregulation, metabolic problems, and a disturbed IL6-mediated acute phase response. The findings provide new insights and therapeutic options in sepsis.

**Keywords** Acute Phase Response Failure; HNF4α Dysfunction; Liver; PPARα-mediated Lipid Dysfunction; Sepsis
**Subject Categories** Immunology; Microbiology, Virology & Host Pathogen Interaction

## Introduction

Based on the evolving knowledge of its mechanisms, sepsis was redefined in 2016 as 'a life-threatening organ dysfunction caused by a dysregulated host response to infection' (Singer et al, 2016). With an annual prevalence of 49 million and 11 million deaths, sepsis treatment remains one of the most unmet medical needs (Rudd et al, 2020). Sepsis progression is characterized by an inflammatory response alternating with hypo-inflammatory phases. This prompted clinical trials to use both pro- and anti-inflammatory drugs but none of them improved survival (Cavaillon and Giamarellos-Bourboulis, 2019). Hence, sepsis therapies in the intensive care unit are limited to antibiotics, vasopressors, fluid resuscitation, and organ support (Evans, 2018). Current sepsis research is focused on aspects beyond inflammation, such as metabolic reprogramming (Van Wyngene et al, 2018).

The pathogenesis of sepsis is characterized by tachycardia, fever, tachypnoea, immune activation, complement and coagulation activation, and the acute phase response (APR), all of which require supraphysiological energy needs. As sepsis patients cannot eat, a starvation response is initiated (Vandewalle and Libert, 2022). Glycogen is rapidly depleted, causing a transient increase in blood glucose levels, followed by proteolysis of muscle proteins and lipolysis of white adipose tissue triglycerides into free fatty acids (FFAs) and glycerol. During normal starvation, plasma FFAs are taken up by the liver (and to some extent by heart and kidney) for oxidation to acetyl-CoA (β-oxidation), which enters the TCA cycle or yields ketone bodies. Amino acids and glycerol are converted to glucose by gluconeogenesis in the liver (and kidney) (Van Wyngene et al, 2018). In sepsis, failure of peroxisome proliferator-activated receptor α (PPARα) and glucocorticoid receptor (GR), which control these processes, causes accumulation of FFAs, glycerol, amino acids and lactate, as well as shortage in ketones and glucose (Vandewalle et al, 2021; Van Wyngene et al, 2020).

The key regulator of FFA catabolism is PPARα (Chamouton and Latruffe, 2012). Downregulation of hepatic PPARα in sepsis at the mRNA and protein levels hampers FFA β-oxidation (Van Wyngene et al, 2020). Consequently, FFAs accumulate in the blood and organs, causing lipotoxicity. Mice treated with the

[1]Center for Inflammation Research, VIB, Ghent, Belgium. [2]Department of Biomedical Molecular Biology, Ghent University, Ghent, Belgium. [3]Institute of Biomedicine, University of Eastern Finland, Kuopio, Finland. [4]Research Group SynBioC, Department of Green Chemistry and Technology, Faculty of Bioscience Engineering, Ghent University, Ghent, Belgium. [5]Experimental Laboratory of Intensive Care, Université Libre de Bruxelles, 1050 Brussels, Belgium. [6]Department of Intensive Care, Center Hospitalier Universitaire de Lille, 59000 Lille, France. [7]Department of Intensive Care Medicine, Ghent University Hospital, Ghent, Belgium. [8]Department of Internal Medicine and Pediatrics, Faculty of Medicine and Health Sciences, Ghent University, Ghent, Belgium. [9]Laboratory of Pharmacology and Toxicology, Department of Pathobiology, Pharmacology and Zoological Medicine, Faculty of Veterinary Medicine, Ghent University, Merelbeke, Belgium. [10]Center for Medical Biotechnology, VIB, Ghent, Belgium. [11]Department of Biomolecular Medicine, Ghent University, Ghent, Belgium. [12]Laboratory of Liver Infectious Diseases, Department of Diagnostic Sciences, Faculty of Medicine and Health Sciences, Ghent University, Ghent, Belgium. [13]Department of Intensive Care, St.-Pierre University Hospital, Université Libre de Bruxelles, 1050 Brussels, Belgium. ✉E-mail: Claude.Libert@irc.vib-ugent.be

PPARα antagonist, GW6471, together with sepsis induced by cecal ligation and puncture (CLP), displayed a significant rise in mortality. Furthermore, the PPARα agonist, pemafibrate, provides protection against CLP by increasing hepatic PPARα expression, thereby reducing lipotoxicity and tissue damage. This underscores the importance of maintaining high PPARα expression levels during CLP. Clinical studies on the use of fibrates in infectious conditions have also been conducted but are still under revision (Tancevski et al, 2014). However, the protective effect of pemafibrate in sepsis is observed only when administered before disease progression (Van Wyngene et al, 2020). This emphasizes the need to understand the mechanism of the PPARα down-regulation. Recent studies support that HNF4α acts as key activator of *Ppara* mRNA expression in the liver by binding to the DR1 motif within its gene promoter (Alder et al, 2014; Pineda Torra et al, 2002). Moreover, HNF4α coactivates several lipid metabolism genes, including *Cpt1*, along with PPARα and Retinoid X receptor-α (RXRα) (Chamouton and Latruffe, 2012; Martinez-Jimenez et al, 2010).

HNF4α, unlike many other nuclear receptors, is constitutively present in the nucleus, where it controls genes involved in the metabolism of lipids, glucose, bile acids and xenobiotics (Tunçer and Banerjee, 2018). HNF4α activity is modulated by alternative splicing, posttranslational modifications, and cofactor interactions. TGFβ is a key upstream regulator in alcoholic hepatitis, inducing *Hnf4a* expression from the P2 promoter (instead of P1) in human hepatocytes (Argemi et al, 2019). Several kinases, including protein kinase A (Viollet et al, 1997), protein kinase C (Sun et al, 2007), ERK1/2 kinase (Vetö et al, 2017), AMPK (Hong et al, 2003), and src kinase (Huck et al, 2019), phosphorylate and inhibit HNF4α. Additionally, CBP-mediated acetylation is essential for the nuclear retention of HNF4α (Soutoglou et al, 2000). Examples of HNF4α cofactors include NCOA1, NCOA2, NCOA3, EP300, CREBBP, GRIP1, and PGC1α.

Due to its critical role in hepatic lineage differentiation and identity, liver architecture is disrupted in hepatocyte-specific HNF4α knockout (Hnf4a^Liver-i-KO) mice, resulting in hepatocellular carcinoma (Ning et al, 2010; Hayhurst et al, 2001). HNF4α dysfunction is associated with multiple liver disorders, such as liver fibrosis and cirrhosis, alcoholic liver disease and non-alcoholic fatty liver disease (NAFLD) (Argemi et al, 2019; Yang et al, 2021; Pan and Zhang, 2022). In this regard, the HNF4α agonist NCT is protective in experimental models of high-fat diet by reducing hepatic steatosis (Lee et al, 2021). However, the role of HNF4α in sepsis and in the regulation of PPARα in sepsis remains unknown.

We report that in the standard CLP model of peritoneal sepsis, hepatic HNF4α progressively loses its transcriptional function due chromatin binding changes. The downstream effects of HNF4α failure in sepsis has a strong effect on PPARα levels and consequent PPARα-mediated FFA consumption, as well as on the expression of several other nuclear receptor TFs in the liver, such as Liver X receptor-α (LXRα), RXRα and Farnesoid X receptor (FXR). HNF4α dysfunction also interferes with interleukin 6 (IL6)-induced, C/EBPβ and STAT3 controlled induction of APR genes and proteins, and hence interferes with liver regeneration in lethal sepsis. Finally, the HNF4α agonist, NCT, reduces sepsis lethality by limiting hepatic steatosis and organ dysfunction and improving hepatic APR.

# Results

## Progressive HNF4α loss-of-function in liver during sepsis

Substantial reprogramming of liver metabolic pathways is a key feature in sepsis (Van Wyngene et al, 2018; Vandewalle and Libert, 2022). To uncover the upstream regulators of these metabolic changes, we performed functional analyses on bulk liver RNA sequencing (RNA-Seq) data collected 8 h and 24 h after sham or CLP (Vandewalle et al, 2021) (Fig. 1A). These time-points reflect early responses to CLP and the acute phase of sepsis, during which the animals are very sick and hypothermic. 2037 genes were differentially expressed 8 h after CLP compared to sham, of which 1001 genes were downregulated (LFC < −1) and 1036 genes were upregulated (LFC > 1) (Fig. 1B; Dataset EV1). After 24 h, gene expression was decreased in 2309 genes and increased in 1635 genes. Focusing on the loss-of-function (LOF) aspects of sepsis, we zoomed in on the genes downregulated after CLP. Most of these genes were found by Enrichr gene list enrichment to be regulated by HNF4α, with a *p*-value of 0.0053 and $2.017 \times 10^{-49}$ at 8 h and 24 h, respectively (Fig. 1C). The difference in *p*-value indicates progressive HNF4α LOF in liver during sepsis. Furthermore, the promoter region of these genes was mainly enriched for the HNF4a motif, with more target promoters containing this motif found at 24 h than at 8 h (511 vs. 219) (Fig. 1D; Dataset EV1). The 219 genes (8 h dataset) functioned in various pathways, such as metabolism of monocarboxylic acid, diacylglycerol, xenobiotics and carbohydrates, while the 511 genes (24 h dataset) were involved in the metabolism of monocarboxylic acid, amino acids and steroids (Fig. 1E). These data indicate that HNF4α dysfunction in sepsis has a broad functional impact.

## Changed intensities and locations of HNF4α in liver chromatin during sepsis

Key aspects of correct HNF4α signaling were investigated to understand the mechanism of HNF4α LOF in septic liver. Although liver *Hnf4a* mRNA was slightly yet significantly downregulated 24 h after CLP but not after 8 h (Fig. 2A), HNF4α protein levels did not change at either timepoint (Fig. 2B,C). In steady-state conditions, HNF4α is constitutively present in the nucleus (Lambert et al, 2020; Yuan et al, 2009). Sepsis did not affect its nuclear presence at several timepoints after CLP. As expected, the HNF4α signal was reduced in liver of Hnf4a^Liver-i-KO mice (Fig. 2D,E).

We evaluated HNF4α DNA binding in the liver by chromatin immunoprecipitation followed by sequencing (ChIP-Seq) 8 h after sepsis onset (Fig. 2F). This timepoint was chosen over the 24 h timepoint to capture the cause rather than the consequence of HNF4α LOF. CLP and sham samples were clearly separated by principal component analysis (PCA), indicating major changes in the HNF4α ChIP-Seq profile during sepsis (Fig. 2G). Genome-wide reduction in HNF4α chromatin binding was observed in septic livers (Fig. 2H). Specifically, we identified about 3500 peaks showing differential signal intensity in sham and CLP, of which 2858 peaks had reduced intensity in CLP (named 'sham-specific' HNF4α binding sites) and 669 peaks had increased intensity in CLP (named 'CLP-specific') (Fig. 2I). The HNF4a motif was enriched for the differential sites, thereby validating the ChIP procedure (Fig. 2J). Other TF motifs within sham-specific HNF4α binding sites were those accommodating PPARa, RARa, RXR, ERRa, COUP-TFII, and

**A.**

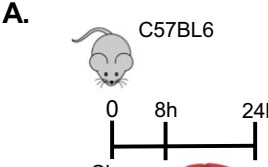

**B.**

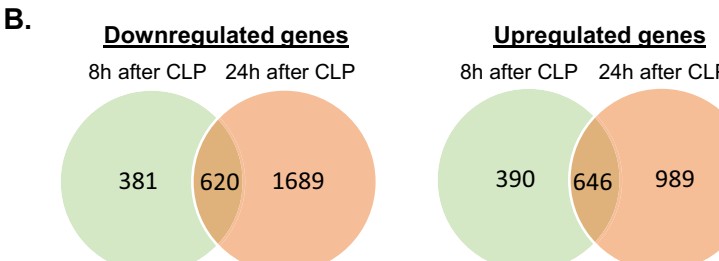

**Downregulated genes**

8h after CLP    24h after CLP

381    620    1689

**Upregulated genes**

8h after CLP    24h after CLP

390    646    989

**C.**

**Enrichr: downregulated genes 8h after CLP**

| | Name | P-value |
|---|---|---|
| 1 | Plagl2 | 0.003892 |
| 2 | Prdm1 (placenta) | 0.004532 |
| 3 | **Hnf4a** | 0.005253 |
| 4 | Prdm1 (small intestine) | 0.01198 |
| 5 | Pou5f1 | 0.01549 |
| 6 | Kdm5a | 0.03098 |

**Enrichr: downregulated genes 24h after CLP**

| | Name | P-value |
|---|---|---|
| 1 | **Hnf4a** | $2.017 \times 10^{-49}$ |
| 2 | Creb1 | $1.021 \times 10^{-12}$ |
| 3 | Glis2 | $5.189 \times 10^{-11}$ |
| 4 | Esrra | $3.356 \times 10^{-10}$ |
| 5 | Plagl2 | $6.117 \times 10^{-10}$ |
| 6 | Nfe2l2 | $5.786 \times 10^{-7}$ |
| 7 | Cdx2 | $6.200 \times 10^{-7}$ |
| 8 | Stat5b | 0.00001583 |
| 9 | Prdm1 | 0.00008977 |
| 10 | Glis3 | 0.02102 |

**D.**

**HOMER motif analysis: downregulated genes 8h after CLP**

| Rank | Motif | Name | P-value | # Target sequences with motif | % Target sequences with motif |
|---|---|---|---|---|---|
| 1 | | **HNF4a** | **$10^{-3}$** | **219** | **21,22%** |
| 2 | | NFY | $10^{-3}$ | 432 | 41,86% |
| 3 | | GFY | $10^{-3}$ | 86 | 8,33% |
| 4 | | HNF1 | $10^{-3}$ | 65 | 6,30% |
| 5 | | Ronin | $10^{-3}$ | 69 | 6,69% |

**HOMER motif analysis: downregulated genes 24h after CLP**

| Rank | Motif | Name | P-value | # Target sequences with motif | % Target sequences with motif |
|---|---|---|---|---|---|
| 1 | | **HNF4a** | **$10^{-8}$** | **511** | **21,90%** |
| 2 | | GFY | $10^{-7}$ | 186 | 7,97% |
| 3 | | HNF1 | $10^{-6}$ | 160 | 6,86% |
| 4 | | Ronin | $10^{-6}$ | 147 | 6,30% |
| 5 | | GFY-Staf | $10^{-5}$ | 159 | 6,82% |

**E.**

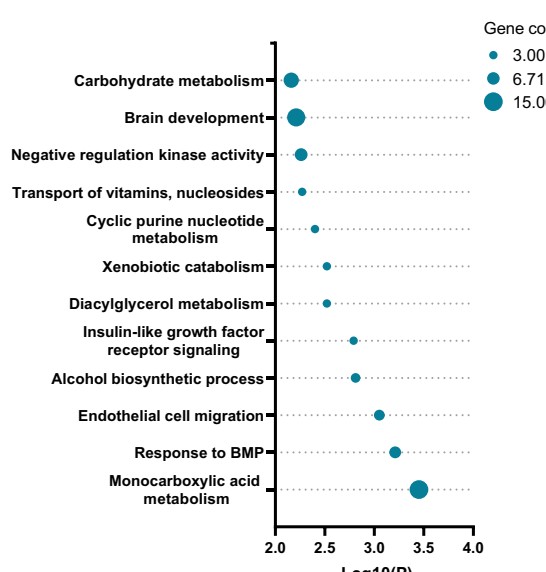

**Metascape pathway analysis: downregulated genes 8h after CLP containing HNF4a motif**

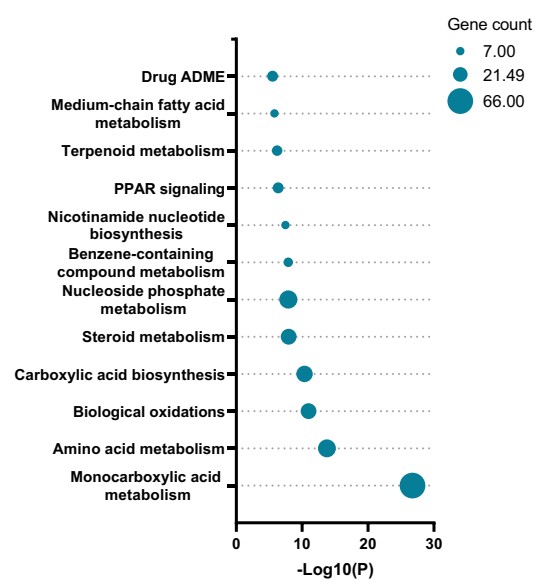

**Metascape pathway analysis: downregulated genes 24h after CLP containing HNF4a motif**

Figure 1. **Progressive HNF4α loss-of-function in liver during sepsis.**

(A) Mice were subjected to sham or CLP and 8 h or 24 h later, liver was isolated for RNA-Seq analysis. $n = 3$/group (biological replicates). (B) Venn diagrams of downregulated (Padj < 0.05, LFC < −1) and upregulated (Padj < 0.05, LFC > 1) genes 8 h and 24 h after CLP. These were used as input for several downstream analyses. *P*-values were calculated from DESeq2 (Wald test). (C) Enrichr analysis (GEO database) of the downregulated genes. The upstream regulators are ordered by their enrichment *p*-value. (D) HOMER motif analysis 1000 bp upstream of the TSS of downregulated genes. (E) Metascape pathway analysis of the downregulated genes containing HNF4a motif 8 h (219 genes) and 24 h (511 genes) after CLP. (C–E) *P*-values derived from Fisher's exact test (Hypergeometric test).

PPARE, while CLP-specific HNF4α binding sites contained CEBP:AP1, NFIL3, HLF, CEBP, ATF4, and ATF1 motifs. 259 of the 2249 genes showing decreased expression (LFC < 0) 8 h after CLP were found near sham-specific HNF4α binding sites (Dataset EV1). Metascape pathway analysis attributed their functions to lipid metabolism, steroid hormone receptor signaling and bile duct development (Appendix Table S1). On the other hand, 106 of the 2560 genes upregulated 8 h after CLP (LFC > 0) were near CLP-specific HNF4α binding sites and were involved in growth factor response, leukocyte homeostasis, apoptosis, regulation of fibroblast migration and bacterial response. At 24 h after CLP, 454 of the 4171 downregulated genes and 137 of the 4188 upregulated genes were found nearby sham-specific and CLP-specific sites, respectively (Dataset EV1). These genes contribute to pathways resembling those of the 8 h dataset (Appendix Table S2). About 16,000 peaks had similar intensities in sham *versus* CLP and were associated with 6482 genes functioning in several metabolic pathways, such as TCA cycle and monocarboxylic acid metabolism (Appendix Fig. S1A,B). Hence, in sepsis, HNF4α LOF is strongly associated with reduced HNF4α chromatin binding in general, and HNF4α seems to be redirected by binding less to metabolism-related genes and more to genes associated with tissue repair.

To investigate the upstream mechanism behind the reduced HNF4α binding in sepsis, we performed an in vitro ELISA-type DNA binding assay on nuclear lysates and examined alternative splicing using in-house paired-end RNA-Seq data 8 h post-CLP (Vandewalle et al, 2021). Of the five transcripts of the mouse *Hnf4a* gene, four of which are protein-coding, only three could be detected in the liver. No significant difference in the HNF4α isoform abundance was observed (Fig. EV1A). Moreover, HNF4α DNA binding was reduced in Hnf4a[Liver-i-KO] samples and in CLP relative to sham 8 h post-CLP (Fig. EV1B,C), suggesting that HNF4α is likely modified in sepsis, potentially through post-translational modifications or altered interactions with cofactors.

## Perturbations in the chromatin binding dynamics of HNF4α during sepsis induce changes in the epigenetic landscape in the liver

To explore the epigenetic alterations underlying the changes in HNF4α chromatin binding during sepsis, we analyzed the liver 8 h after sham or CLP with 'Assay for transposase-accessible chromatin using sequencing' (ATAC-Seq), as well as ChIP-Seq for H3K4me3 and H3K27ac (Fig. 3A). H3K4me3 is a hallmark of active promoters, whereas H3K27ac is a marker for active enhancers. ATAC-Seq and H3K27ac ChIP-Seq analyses showed clear separation in a PCA plot, indicating notable changes in their profiles during sepsis (Figs. 3B and EV2A). Indeed, at the genome-wide level, chromatin accessibility in the liver was reduced in CLP (Fig. 3C). About 20,000 chromatin sites were remodeled, of which

13,641 were closing ('sham-specific' open chromatin sites) and 6223 were opening ('CLP-specific') (Fig. 3D). CLP had minimal effect on H3K4me3, but a significant effect on H3K27ac, with 4864 enhancer regions showing decreased acetylation ('sham-specific') and 3754 enhancer regions showing increased acetylation ('CLP-specific'). We further characterized the differential ATAC-Seq sites by HOMER motif analysis and Metascape pathway analysis of the nearby genes. Consistent with the findings of HNF4α ChIP-Seq analysis, sham-specific open chromatin sites contained motifs associated with PPARa, COUP-TFII, RXR, and PPARE (Fig. EV2B). Conversely, motifs representing CEBP, CEBP:AP1, NFIL3, HLF, and Fos were enriched within CLP-specific open chromatin sites. Interestingly, sites that were closing in CLP were highly enriched in the HNF4a motif. Of 2249 genes downregulated 8 h after CLP (LFC < 0), 896 were nearby sham-specific open chromatin sites (Dataset EV1). These were involved in lipid metabolism, monocarboxylic acid metabolism, and regulation of GTPase activity (Appendix Table S3). In contrast, 617 of the 2560 genes upregulated 8 h after CLP (LFC > 0) were near CLP-specific open chromatin sites and functioned in Egfr1 signaling, cytokine signaling, regulation of epithelial cell migration, and regulation of programmed cell death. For 24 h after CLP, 1529 of the 4171 downregulated genes and 831 of the 4188 upregulated genes were found nearby sham-specific and CLP-specific sites, respectively (Dataset EV1). These genes functioned in similar pathways as those in the 8 h dataset (Appendix Table S4). Moreover, 561 of the 2560 and 826 of the 4188 genes upregulated 8 h and 24 h after CLP, respectively, were nearby sham-specific open chromatin sites, while 217 of the 2249 and 422 of the 4171 genes downregulated 8 h and 24 h after CLP, respectively, were found near CLP-specific open chromatin sites (Dataset EV1). About 25,000 peaks had similar intensities in sham and CLP and were associated with 6364 genes functioning in several metabolic pathways, such as TCA cycle and carbohydrate biosynthesis (Appendix Fig. S1C,D).

To characterize the chromatin structure and function at the differential HNF4α binding sites in CLP, we plotted the intensities of the ATAC, H3K4me3 and H3K27ac signals found in CLP at these sites. Regions displaying diminished/enhanced HNF4α chromatin binding exhibited only a tendency to reduced/enhanced accessibility, respectively (Fig. 3E). In contrast, the H3K27ac state was significantly altered in these regions during CLP, with decreased acetylation at sham-specific sites and increased acetylation at CLP-specific sites (Figs. 3F and EV2C). Additionally, we used public data (Qu et al, 2021) to plot the intensity of the H3K27ac signal found in Hnf4a[Liver-i-KO] at the differential HNF4α binding sites in CLP. Both sham and CLP-specific sites showed significantly less H3K27 acetylation in the KO (Figs. 3F and EV2C). Similarly, we compared the ATAC-Seq data from CLP with publicly available data from Hnf4a[Liver-i-KO] samples (Hunter et al, 2022). Sham-specific open chromatin sites were significantly less accessible in the KO, while CLP-specific open

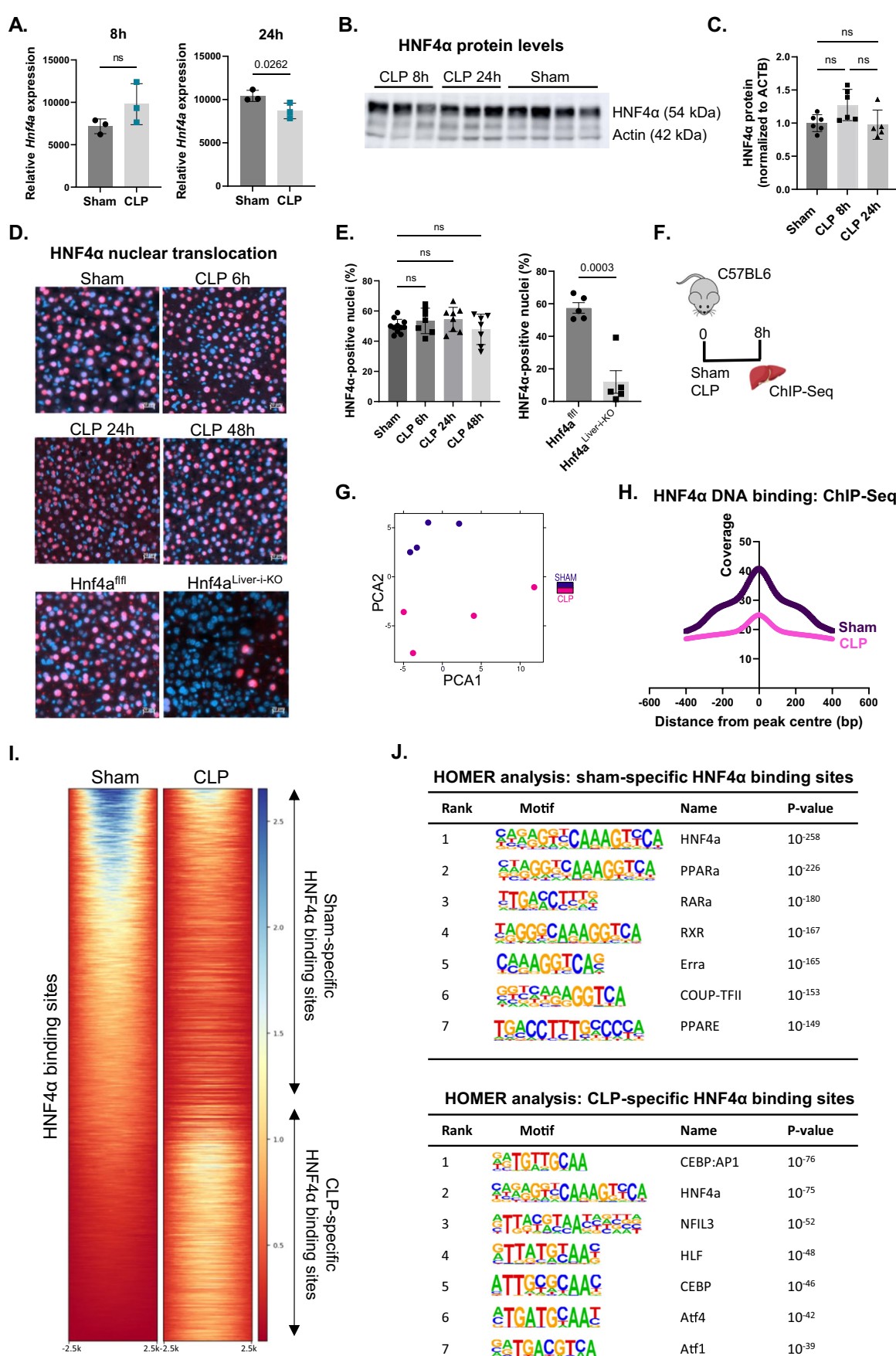

**Figure 2. Changed intensities and locations of HNF4α in liver chromatin during sepsis.**

(A) Normalized counts of *Hnf4a* in liver 8 h and 24 h after CLP (RNA-Seq data). $n = 3$/group. (B, C) Western blot analysis of HNF4α protein levels in liver 8 h and 24 h after CLP or sham. HNF4α protein expression (54 kDa) was normalized to actin levels (42 kDa) and expressed relative to sham. $n = 5–6$/group. (D, E) Immunofluorescent images of liver 6 h, 24 h and 48 h after CLP or sham, or from tamoxifen-injected Hnf4a^flfl and Hnf4a^Liver-i-KO mice, stained with HNF4α antibody (red) and DAPI (blue). Scale bar = 20 μm. Percentage HNF4α positive nuclei was quantified. $n = 5–11$/group. (F–J) HNF4α ChIP-Seq in liver 8 h after sham or CLP. $n = 4$/group. (F) Experimental setup. (G) PCA plot. (H) Signal intensity as fold over local region per position in a 400-bp region centered on the center of all called peaks. (I) Heatmap representing the region scores by deepTools 2.5 kbp centered on the center of all differential peaks (Padj < 0.05). P-values were calculated from DESeq2 (Wald test). Scores were derived from bam file coverage with deepTools. Higher scores indicate higher coverage. (J) HOMER transcription factor motif enrichment 200 bp centered on the center of all differential peaks. P-values derived from Fisher's exact test (Hypergeometric test). Bars: mean ± SD, except for (E) mean ± SEM. Each dot represents a single biological replicate. P-values were analyzed with unpaired t-test (A, E) or one-way ANOVA (C, E). ns: non-significant. Source data are available online for this figure.

chromatin sites remained unaffected by hepatic HNF4α depletion (Fig. 3G). Altogether, CLP is marked by chromatin remodeling in the liver, as well as changes in H3K27 acetylation, with a relatively lesser impact on H3K4 methylation. The data indicate that changes in HNF4α chromatin binding in the liver during sepsis affect mainly H3K27 acetylation, thereby modulating enhancer activity, with little effect on chromatin accessibility.

## Increased sepsis lethality and metabolic dysfunction in Hnf4a^Liver-i-KO mice

We generated Hnf4a^Liver-i-KO mice to gain a better understanding of the downstream consequences of HNF4α LOF in liver during sepsis progression. Bulk liver analyses (RNA-Seq, HNF4α ChIP-Seq, and ATAC-Seq) showed that lipid metabolism was the common pathway inhibited by CLP downstream of HNF4α (Fig. 4A). As HNF4α is essential for maintaining hepatocyte identity, we conducted our experiments 3 days after the last tamoxifen injection, in HNF4α-deficient conditions, as confirmed by Western blot analysis (Fig. 4B), but before liver de-differentiation was complete. Hnf4a^Liver-i-KO mice were significantly more sensitive to CLP than control mice (Hnf4a^flfl) (Fig. 4C), which emphasizes the critical role of HNF4α in sepsis. We performed bulk liver RNA-Seq 8 h after sham or CLP in Hnf4a^flfl and Hnf4a^Liver-i-KO mice (Fig. 4D). Then, we compared gene expression changes between those downregulated in CLP relative to sham in Hnf4a^flfl as well as in Hnf4a^Liver-i-KO relative to Hnf4a^flfl in sham (Fig. 4E left panel; Dataset EV1). These genes are considered constituents of protective pathways that weaken in sepsis due to hepatic HNF4α LOF but can diminish sepsis lethality when activated/overexpressed. Examples include *Ppara* (Van Wyngene et al, 2020), *Nr1h3* (Wang et al, 2011a), *Cyp7a1* (Liu et al, 2016) and *Rxr* (Dolin et al, 2023) (Fig. 4F). Metascape pathway analysis and Enrichr suggest their involvement in bile acid metabolism, steroid biosynthesis, and lipid metabolism, particularly triglyceride hydrolysis (*Daglb*, *Smpd3*), FFA oxidation (*Ppara*) and VLDL secretion (*Acsl3*, *Ldlr*) (Figs. 4E and EV3 left panel). On the other hand, genes upregulated in Hnf4a^flfl by CLP as well as in Hnf4a^Liver-i-KO relative to Hnf4a^flfl in sham may be regarded as components of toxic pathways that become more active during sepsis due to the HNF4α LOF (Fig. 4E right panel; Dataset EV1). These genes are potential targets for inhibitors to reduce sepsis lethality. Examples include *Pfkfb3* (Xiao et al, 2023), *S100g* (Yao et al, 2021), *Ptges* (Gurusamy et al, 2021), and *Pdk4* (Mainali et al, 2021) (Fig. 4F). Overall, the genes play roles in transcriptional regulation and cell death pathways (Fig. 4E right panel). They are also involved in hypoxia, TNFα signaling, apoptosis, and mTOR signaling, all of which have been implicated in sepsis progression (Zhang and Ning, 2021) (Fig. EV3 right panel).

To elucidate the pathways contributing to the increased sepsis lethality in Hnf4a^Liver-i-KO mice, we also identified the genes downregulated and upregulated in CLP in the absence of HNF4α in hepatocytes (Fig. 4G). The more strongly downregulated genes functioned mainly in lipid metabolism, particularly FFA oxidation (*Ppara*, *Thrb*, *Foxa2*) and VLDL secretion (*Acsl3*, *Ldlr*), while the more strongly upregulated genes contributed to cell cycle, inflammation and cell death pathways. As examples, Fig. 4H shows the expression of lipid metabolic genes *Acsl3*, *Ppara* and *Ldlr* (further downregulated in Hnf4a^Liver-i-KO CLP), cell cycle genes *Chek1* and *Bmyc* and inflammatory gene *Il20rb* (further upregulated in Hnf4a^Liver-i-KO CLP). To summarize, depleting HNF4α enhances sepsis sensitivity characterized (transcriptionally) by increased lipid dysfunction, inflammation and cell death. Hence, HNF4α LOF may contribute to problematic FFA oxidation and VLDL secretion, perturbations in bile acid metabolism and cholesterol homeostasis, as well as the inflammatory cascade and cellular damage frequently encountered in sepsis.

## Hepatic HNF4α loss-of-function in sepsis impairs PPARα expression, aggravating lipid metabolic dysfunction

In septic mice and pigs, reduced hepatic PPARα levels compromise the uptake and oxidation of lipids, which accumulate in the blood and liver. This leads to suboptimal ATP and ketone production (Van Wyngene et al, 2020; Vandewalle et al, 2022). HNF4αKO RNA-Seq data from liver already suggested HNF4α failure upstream of FFA oxidation and VLDL secretion problems in sepsis (Fig. 4E–H). Besides *Ppara*, several other important nuclear receptors were downregulated in CLP, as well as in Hnf4a^Liver-i-KO sham, and were further reduced in Hnf4a^Liver-i-KO CLP, relative to Hnf4a^flfl CLP (Fig. 5A). These include *Rxra*, *Nr1h4* (FXR), *Nr1i3* (Constitutive Androstane Receptor, CAR), *Ar* (Androgen Receptor) and *Nr1h3* (LXRα). Conversely, the expression of other nuclear receptors, such as *Nr3c2* (encoding Mineralocorticoid Receptor, MR), *Nr0b2* (encoding Small Heterodimer Partner, SHP) and *Esrra*, were decreased in CLP but elevated in Hnf4a^Liver-i-KO sham and showed less reduction in Hnf4a^Liver-i-KO CLP. The diminished expression of PPARα in Hnf4a^Liver-i-KO mice following CLP was confirmed at both the RNA and protein levels (Fig. 5B,C). In normal conditions, HNF4α induces *Ppara* expression by binding directly to the DR1 element within the *Ppara* promoter (Alder et al, 2014; Pineda Torra et al, 2002). However, ChIP-qPCR analysis demonstrated a diminished association of HNF4α with the promoter region of *Ppara* during sepsis (Fig. 5D). Furthermore,

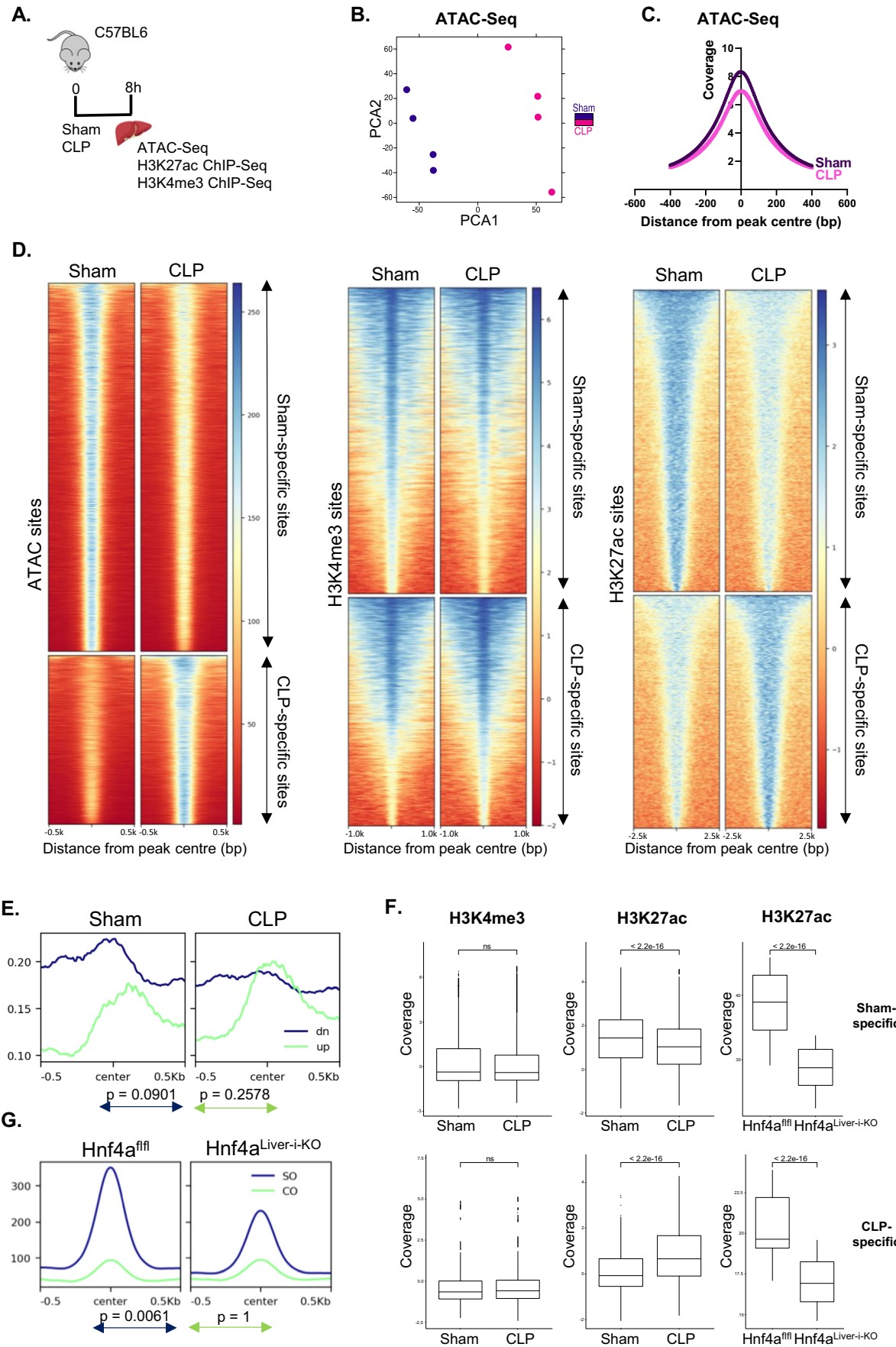

◀

**Figure 3. Perturbations in the chromatin binding dynamics of HNF4α during sepsis induce changes in the epigenetic landscape in the liver.**

(A) Experimental setup. Liver was isolated 8 h after sham or CLP for ATAC-Seq, and H3K27ac and H3K4me3 ChIP-Seq analyses. $n = 4$/group (biological replicates). (B) ATAC-Seq PCA plot. (C) Signal intensity as fold over local region per position in a 400-bp region centered on the center of all called ATAC-Seq peaks. (D) Heatmaps representing the region scores by deepTools centered on the center of all differential ATAC-Seq, and H3K4me3 and H3K27ac ChIP-Seq peaks (Padj < 0.05). $P$-values were calculated from DESeq2 (Wald test). Higher score indicates higher coverage. (E) Sham and CLP ATAC signals 0.5 kbp centered on the center of HNF4α ChIP-Seq peaks with decreased intensity (dn-blue) and increased intensity (up-green) in CLP. (F) Boxplots representing H3K27ac and H3K4me3 signals from sham and CLP or Hnf4a$^{fl/fl}$ and Hnf4a$^{Liver-i-KO}$ 1.5 kbp or 1.0 kbp, respectively, centered on the center of HNF4α ChIP-Seq peaks with decreased intensity ('sham-specific') ($n = 2858$) and increased intensity ('CLP-specific') ($n = 669$) in CLP. The boxplots display the distribution from Q1 (the first quartile, 25%) to Q3 (the third quartile, 75%), where the box itself spans the interquartile range (IQR). The line inside the box indicates the median. The whiskers extend from $Q1 - 1.5 * IQR$ to $Q3 + 1.5 * IQR$. Outliers are defined as values that fall outside this range, calculated using the formulas: $\min(\max(values), Q\_3 + 1.5 * IQR)$ for the upper bound and $\max(\min(values), Q\_1 – 1.5 * IQR)$ for the lower bound. (G) Hnf4a$^{fl/fl}$ and Hnf4a$^{Liver-i-KO}$ ATAC signals 0.5 kbp centered on the center of sham-specific (SO-blue) and CLP-specific (CO-green) ATAC-Seq peaks. (E–G) $P$-values were analyzed with Wilcoxon test. ns: non-significant.

PPARα biological activity was studied in the liver of Hnf4a$^{Liver-i-KO}$ mice by qPCR of PPARα dependent genes 4 h after GW7647 (PPARα agonist) administration (Fig. 5E). PPARα target genes such as *Hmgcs2*, *Slc25a20,* and *Acox1*, including *Ppara* itself, were unresponsive to GW7647 in Hnf4a$^{Liver-i-KO}$ mice, indicating loss of PPARα activity in the absence of HNF4α (Fig. 5F). These data indicate that HNF4α may be directly involved in gene regulation (*Ppara* gene), as well as indirectly via transactivation of PPARα in the cases of *Hmgcs2*, *Slc25a20*, and *Acox1*.

Several lipid metabolic parameters were measured 8 h and 24 h after CLP or sham in Hnf4a$^{Liver-i-KO}$ mice (Fig. 5G). Plasma FFA levels were slightly increased 8 h after CLP and were significantly upregulated further in Hnf4a$^{Liver-i-KO}$ CLP (Fig. 5H). No difference in basal levels could be observed, which might be due to FFA uptake by the liver. However, when hepatic FFA β-oxidation fails and VLDL secretion is reduced, as in Hnf4a$^{Liver-i-KO}$ mice, FFAs in the liver are included in lipid droplets (Van Wyngene et al, 2020; Heeren and Scheja, 2021). We therefore performed LipidTox staining to visualize these droplets 8 h after CLP (Appendix Fig. S2A). Indeed, the total volume of lipid droplets, expressed as the sum of voxel counts, was significantly increased in the liver in the absence of HNF4α (Fig. 5I). However, this increase was not augmented by CLP. Moreover, Hnf4a$^{Liver-i-KO}$ mice, relative to Hnf4a$^{fl/fl}$ mice, tended to have higher plasma IL6 levels 24 h after CLP, indicating a more pro-inflammatory environment (Fig. 5J). Furthermore, organ dysfunction was more pronounced in Hnf4a$^{Liver-i-KO}$ mice 24 h after CLP, reflected in higher plasma aspartate aminotransferase (AST) and lactate dehydrogenase (LDH) levels (Fig. 5K). Plasma alanine aminotransferase (ALT) levels were increased in only part of the CLP mice (Appendix Fig. S2B). AST levels typically surpass ALT levels, because the former involve not only liver damage, but also cardiac and skeletal muscle damage. Altogether, HNF4α LOF in sepsis downregulates the expression of many nuclear receptors, including PPARα. Furthermore, HNF4α failure aggravates lipid dysfunction and inflammation, which may contribute to aggravating organ damage.

## Hepatic HNF4α loss-of-function results in reduced IL6-mediated C/EBPβ and STAT3 controlled acute phase response in lethal sepsis

We hypothesized that HNF4α is redirected in sepsis by binding less to metabolism-related genes and more to genes associated with tissue repair. We found that the promoters of the latter genes were mainly enriched for the CEBP motif, followed by NFIL3 and HLF motifs

(Figs. 2J and EV2B). C/EBPβ is essential in the synthesis of acute phase proteins (APPs) in hepatocytes in response to IL6 released from leukocytes during infection or injury (Alonzi et al, 2001; Goldstein et al, 2017). Because IL6-mediated C/EBPβ and STAT3 controlled acute phase response (APR) plays a crucial role in host defense and liver regeneration (Moshage, 1997), we studied the role of HNF4α in the IL6-mediated response. Bulk liver RNA-Seq was performed 3 h after IL6 or PBS injection in Hnf4a$^{fl/fl}$ and Hnf4a$^{Liver-i-KO}$ mice (Fig. 6A). Contrary to the repurposing hypothesis of HNF4α, genes significantly induced by IL6 in Hnf4a$^{fl/fl}$ mice were generally 1.5x less upregulated in Hnf4a$^{Liver-i-KO}$ mice, indicating a reduced IL6 response in the absence of HNF4α in hepatocytes (Fig. 6B). Some IL6-responsive genes were upregulated in Hnf4a$^{Liver-i-KO}$ compared to Hnf4a$^{fl/fl}$ mice basally, and their expression was further enhanced by IL6, but most IL6 target genes were downregulated by HNF4α depletion and responded less to IL6 injection (Fig. 6C). The latter group, without exception, contained 21 genes involved in the APR (Appendix Table S5). For example, the expression of *Saa1* and *Apcs*, the two most important acute phase genes in mice, is shown in Fig. 6D. Besides the APR, IL6-activated, HNF4α-dependent genes functioned in death receptor signaling and LXR/RXR activation (Appendix Table S5). On the other hand, genes activated by IL6 and upregulated by HNF4α depletion belonged more to general transcription and cell cycle-related pathways. Moreover, C/EBPβ and STAT3 were the top upstream regulators controlling the IL6 target genes downregulated in Hnf4a$^{Liver-i-KO}$ mice (Fig. 6E,F). *Stat3* itself was slightly less upregulated by IL6 in the absence of HNF4α (Fig. EV4A). Furthermore, STAT3 activation by Tyr705 phosphorylation, measured as the ratio of pSTAT3 to total STAT3, was significantly decreased in the liver of Hnf4a$^{Liver-i-KO}$ mice 8 h after CLP (Fig. EV4B–D). This strongly suggests that HNF4α is required to support sufficient STAT3 activity in the liver during sepsis. ELISA of APPs serum amyloid A (SAA) and serum amyloid P (SAP) further confirmed the lack of APR in the absence of HNF4α (Fig. 6G).

Compensatory hepatic regeneration has been observed after mild, intraperitoneal sepsis in rats (Weiss et al, 2001). We therefore reasoned that sepsis lethality is determined by metabolic dysfunction and the ability to induce an APR in response to IL6. To study this, HNF4α-dependent IL6 target genes involved in APR (from RNA-Seq) were measured in sublethal and lethal CLP, by which IL6 specificity was determined by IL6 injection 24 h after sham or CLP (Fig. 6H). Body temperature (BT) was used to distinguish sublethal from lethal CLP (Fig. 6I). BT initially dropped in sublethal CLP mice but recovered 24 h after CLP, while the BT of lethal CLP mice continued to drop over time. We found that *Apcs* was significantly downregulated in lethal compared to sublethal

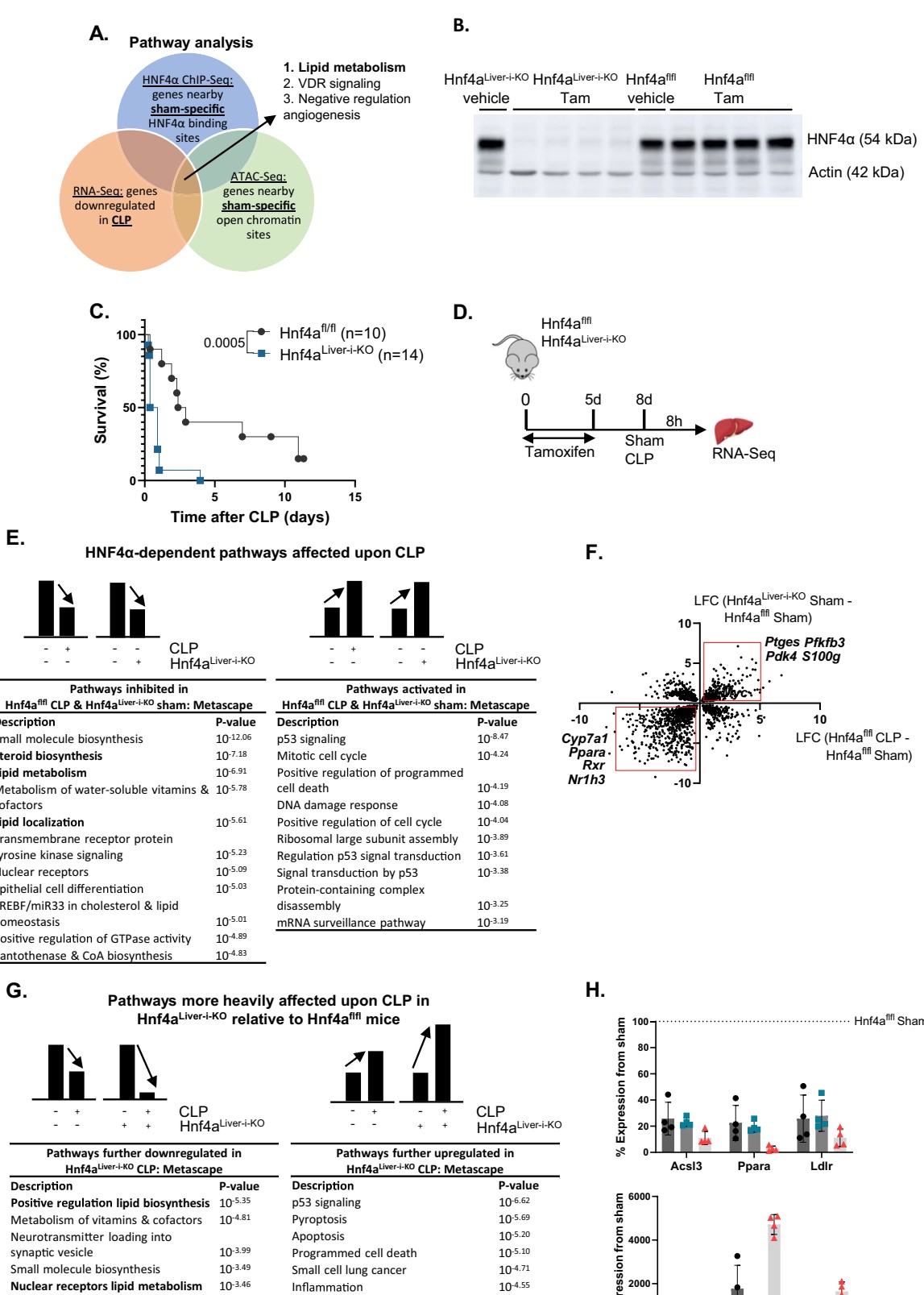

◀ **Figure 4. Increased sepsis lethality and metabolic dysfunction in Hnf4a^Liver-i-KO mice.**

(A) Overlap between RNA-Seq, HNF4α ChIP-Seq and ATAC-Seq-associated pathway analysis for genes with Padj < 0.05 & LFC < 0 in liver 8 h after sham or CLP. *P*-values were calculated from DESeq2 (Wald test). RNA-Seq: *n* = 3/group; ChIP-Seq & ATAC-Seq: *n* = 4/group (biological replicates). (B) Western blot analysis of hepatic HNF4α protein levels (54 kDa) relative to actin (42 kDa) in Hnf4a^Liver-i-KO and Hnf4a^fl/fl mice 3 days after tamoxifen or vehicle treatment. *n* = 12 (4 female, 8 male). (C) Hnf4a^Liver-i-KO and Hnf4a^fl/fl mice were subjected to CLP 3 days after the last tamoxifen injection and mortality was monitored over 10 days. Data were analyzed with a Log-rank test. *n* = 24 (11 female, 13 male). (D) Hnf4a^Liver-i-KO and Hnf4a^fl/fl mice were *i.p.* injected with tamoxifen for 5 consecutive days. Three days later, sham or CLP was performed, and liver was isolated for RNA-Seq analysis 8 h later. *n* = 4/group (9 female, 7 male). (E) Metascape pathway analysis of genes downregulated or upregulated (Padj < 0.05) in Hnf4a^Liver-i-KO sham relative to Hnf4a^fl/fl, and downregulated (Padj < 0.05, LFC < −0.8) (*n* = 716) or upregulated (Padj < 0.05, LFC > 0.8) (*n* = 245), respectively, in CLP relative to sham. (F) Scatterplot representing LFC of genes with Padj < 0.05 in Hnf4a^Liver-i-KO sham relative to Hnf4a^fl/fl sham, and in Hnf4a^fl/fl CLP relative to Hnf4a^fl/fl sham. Key genes of interest are indicated. (G) Metascape pathway analysis of genes further downregulated (Padj < 0.05, LFC < −0.8) (*n* = 288) or upregulated (Padj < 0.05, LFC > 0.8) (*n* = 133) in Hnf4a^Liver-i-KO CLP, relative to Hnf4a^fl/fl CLP. (H) Normalized counts of representative genes from (G), with the expression level of Hnf4a^fl/fl sham set as 100%. *n* = 4/group. Bars: mean ± SD. Each dot represents a single biological replicate. (E, G): *P*-values derived from Fisher's exact test (Hypergeometric test). (F, H): *P*-values were calculated from DESeq2 (Wald test). Source data are available online for this figure.

CLP, despite increasing plasma IL6 levels (Fig. 6J,K). The same trend was observed for other acute phase genes, such as *Saa1*, *Saa2*, *A2m*, and *Hpx* (Fig. EV4E). IL6 injection in these mice 24 h after sham or CLP confirmed the absence of IL6-induced *Apcs* expression in CLP (shown as fold induction) (Fig. 6L). SAP protein, encoded by *Apcs*, also tended to be lower in lethal CLP (Fig. 6M). Furthermore, IL6 was less able to activate STAT3 in lethal compared to sublethal CLP (Fig. EV4F,G). To test our hypothesis that induction of a proper APR partly determines sepsis sensitivity, we treated mice with IL6 and DEX 12 h before CLP. It is well documented that co-administration of IL6 and glucocorticoids activates the APR in hepatocytes (Dittrich et al, 2012). Indeed, mice treated with the combination of IL6 and DEX showed a significant increase in plasma SAA levels and were partially protected against CLP in a HNF4α-dependent manner (Fig. 6N,O). In summary, despite increasing plasma IL6 levels, IL6 fails to induce a proper APR in lethal CLP. This failure, characterized by less STAT3 phosphorylation and less APP production mediated by C/EBPβ and STAT3, can be attributed to the HNF4α LOF in sepsis and contributes to the lethality of CLP.

## HNF4α agonist NCT partially protects against polymicrobial sepsis by reducing hepatic steatosis and improving hepatic acute phase response

A recent high-throughput screening for new HNF4α agonists discovered N-trans-caffeoyltyramine (NCT) as a new, potent and specific activator of HNF4α (Lee et al, 2021). Its structure is similar to the (weak) HNF4α agonists benfluorex and alverine (Lee et al, 2013) (Fig. 7A). NCT was introduced as a potential NAFLD drug candidate due to its ability to recover HNF4α expression and reduce hepatic steatosis (Lee et al, 2021; Veeriah et al, 2022). In light of the similarities between NAFLD and sepsis regarding hepatic lipid dysfunction, we investigated the efficacy of NCT in the CLP model. Whether administered prophylactically 7 days before CLP induction or therapeutically, with one dose 3 h post-CLP and another 24 h post-CLP, the mice exhibited partial yet significant protection against death (Fig. 7B,C). As mentioned above, hepatic HNF4α LOF during sepsis, attributed to reduced chromatin binding, has implications for lipid metabolism and IL6-mediated APR downstream. To assess whether NCT can ameliorate these effects, mice pre-treated with NCT underwent either sham or CLP surgery (Fig. 7D). NCT treatment conferred protection at BT level 24 h post-CLP, regardless of the bacterial load in the blood and liver

(Fig. 7E,F). This underscores the involvement of HNF4α in tolerance to sepsis. CLP significantly reduced HNF4α chromatin binding to the promoters of various target genes, including *Ppara*, and decreased expression of *Ppara* in the liver (Fig. 7G,H). However, this decrease was absent when NCT was pre-administered to mice. During sepsis, NCT also reduced FFA accumulation in the blood and liver (steatosis) (Fig. 7I,J; Appendix Fig. S3A). Besides its protection at the metabolic level, NCT also upregulated the expression of acute phase gene *Apcs* in the liver 24 h after CLP, reflected by increased SAP protein in the plasma (Fig. 7K,L). The same trend was observed for other acute phase genes, such as *Saa1*, *Saa2*, *A2m*, and *Hpx* (Appendix Fig. S3B). The effect of NCT on acute phase gene expression occurred regardless of the IL6 dose, which was even decreased in CLP after NCT (Fig. 7M). We believe that low-dose IL6 is necessary for its role in regeneration, while high-dose IL6 contributes to inflammation (Deutschman et al, 2006; Riedemann et al, 2003). NCT hereby creates a more controlled inflammatory environment in sepsis characterized by less organ damage, as measured by plasma AST, ALT, and LDH levels (Fig. 7N; Appendix Fig. S3C). These data collectively suggest that targeting HNF4α in mice can partially protect against CLP-induced peritoneal sepsis. Moreover, NCT pretreatment reduces sepsis lethality by (1) inducing *Ppara* expression and thereby limiting lipid accumulation in the blood and liver, (2) increasing acute phase gene and protein expression, which is required for proper liver regeneration, and (3) limiting the IL6 dose and thereby reducing systemic inflammation.

## The relevance of HNF4α loss-of-function in porcine sepsis and in septic mice with humanized liver

In CLP mice, we observed a gradual decline in the functionality of HNF4α, leading to significant effects on hepatic metabolism and the APR. To extend the relevance of these findings beyond mice, we employed a porcine (fecal installation) sepsis and a humanized liver mouse CLP model. In the pig model, septic shock was induced by intraperitoneal administration of autologous feces, resulting in fecal peritonitis (Vandewalle et al, 2022). Livers were collected for bulk RNA-Seq 18 h after sepsis onset (Fig, EV5A). Like CLP mice, analyses using Enrichr and HOMER motifs showed that most of the genes downregulated by sepsis in pigs are controlled by HNF4α (Fig. EV5B,C). In the humanized liver mouse model, severe combined immunodeficiency (SCID) mice carrying the Alb-uPA transgene (uPA^+/+) were engrafted with human hepatocytes to

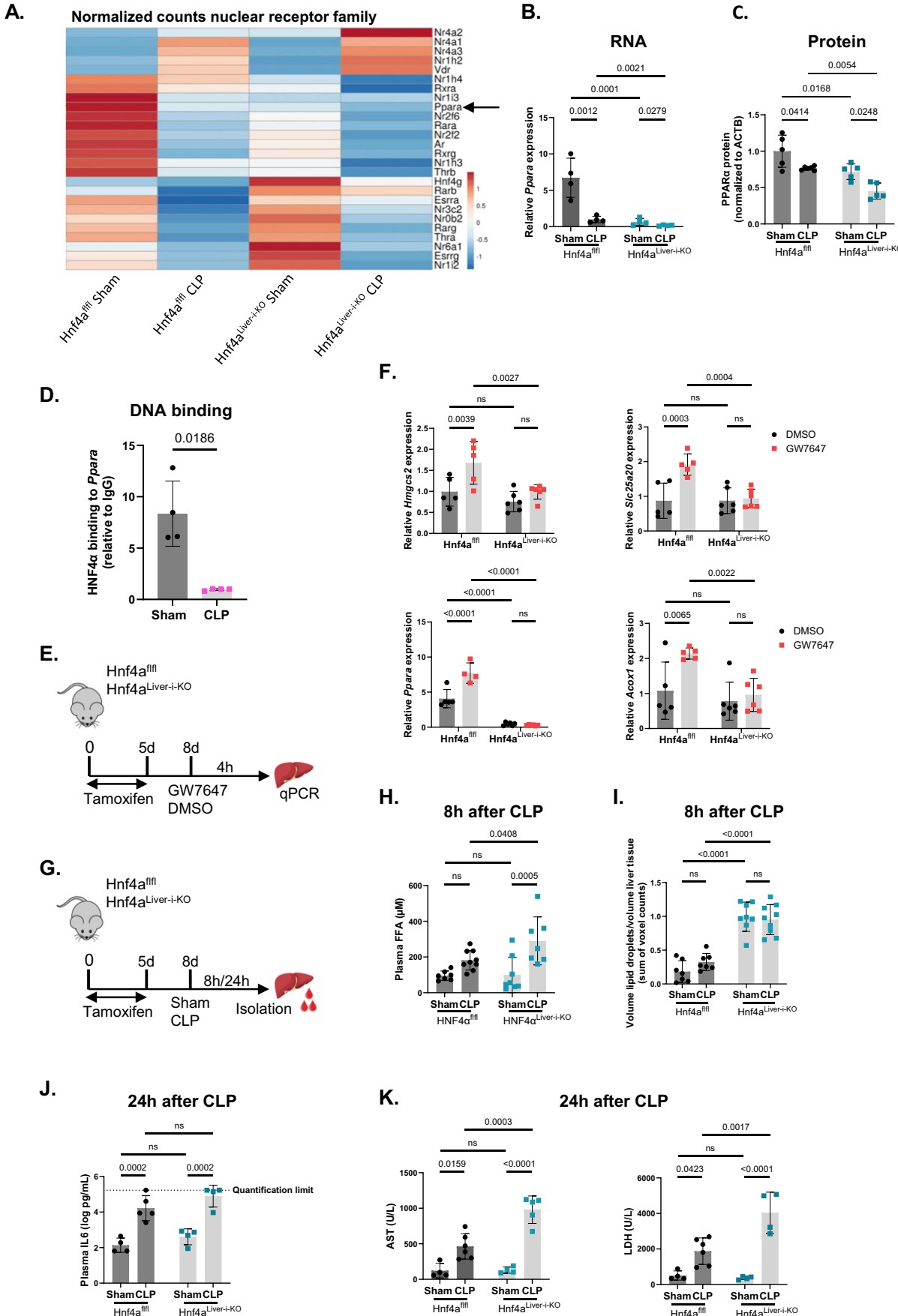

**Figure 5. Hepatic HNF4α loss-of-function in sepsis impairs PPARα expression, aggravating lipid metabolic dysfunction.**

(A–C) Analysis of livers from tamoxifen-injected Hnf4a^Liver-i-KO and Hnf4a^fl/fl mice 8 h after sham or CLP. (A) Heatmap of differentially expressed nuclear receptor genes from RNA-Seq (unit scale bar: normalized counts). Rows are centered by unit variance scaling and clustered using correlation distance and average linkage. n = 4/group (9 female, 7 male) (biological replicates). (B) RT-qPCR *Ppara* mRNA expression relative to *Gapdh* and *Hprt*. (C) Western blot analysis of PPARα protein levels (52 kDa) relative to actin (42 kDa). n = 5/group. (D) HNF4α ChIP-qPCR on *Ppara* promoter in liver 8 h after sham or CLP. n = 4/group. (E) PPARα agonist GW7647 or vehicle were injected *i.p.* in Hnf4a^Liver-i-KO and Hnf4a^fl/fl mice 3 days after tamoxifen treatment. Livers were isolated for qPCR 4 h later. n = 5–6/group (2 female, 20 male). (F) RT-qPCR mRNA expression of *Ppara* and several PPARα target genes relative to *Rpl* and *Hprt*. (G) Hnf4a^Liver-i-KO and Hnf4a^fl/fl mice were *i.p.* injected with tamoxifen on 5 consecutive days. Three days later, sham or CLP was performed, and liver and blood were isolated 8 h and 24 h later. (H) Plasma FFA levels 8 h after sham or CLP. n = 31 (12 female, 19 male). (I) Volume of lipid droplets (represented by voxel counts), relative to liver tissue volume, was calculated for each z-stack and averaged over all z-stacks per section 8 h after sham or CLP. n = 31 (13 female, 18 male). (J, K) Plasma IL6, aspartate aminotransferase (AST) and lactate dehydrogenase (LDH) levels 24 h after sham or CLP. n = 17–19 (7 female, 10–12 male). Bars: mean ± SD. Each dot represents a single biological replicate. *P*-values were analyzed with two-way ANOVA (B, C, F, H–K) or unpaired t-test (D). ns: non-significant. Source data are available online for this figure.

replace the killed mouse hepatocytes (Meuleman et al, 2005). The repopulation of human hepatocytes in the liver was determined by the ratio of human to mouse albumin in the blood of the mice, which was 30% in our case. The 70% mouse hepatocytes express the uPA transgene but derive from constant proliferation and progenitor cell differentiation. Subsequently, the mice were subjected to CLP or sham procedures, and livers were collected for qPCR 24 h after sepsis onset (Fig. EV5D). To investigate HNF4α LOF in these mice, we analyzed by RT-qPCR with human-specific and mouse-specific primers, the expression of several HNF4α-dependent genes that significantly decreased in wild-type mice 24 h post-CLP. This examination extended to mice, pigs, and humanized mice, revealing consistency in all three sepsis models (Fig. EV5E).

# Discussion

FOXA2 and HNF4α are the key regulators of hepatocyte identity (Tunçer and Banerjee, 2018). They mainly control functional differentiation, while YAP and TAZ regulate more the quantitative aspects of de-differentiation and regeneration (Moya and Halder, 2019; Alder et al, 2014). During liver regeneration, HNF4α is downregulated to allow de-differentiation and proliferation, followed by its subsequent re-expression to restore function and terminate regeneration (Huck et al, 2019). Our data identify an HNF4α loss-of-function (LOF) in mouse liver during sepsis, as observed in chronic liver diseases such as NAFLD and alcoholic hepatitis (AH) (Pan and Zhang, 2022; Argemi et al, 2019). We propose that this LOF represents a physiological 'reflex' of the liver to switch to de-differentiation and regeneration. However, in chronic conditions and in sepsis, this switch may be detrimental due to the persistent downregulation of hepatocyte identity genes leading to permanent liver dysfunction. Here, the HNF4α LOF was not limited to the mouse CLP model, but was also observed in porcine sepsis and in human hepatocytes in a humanized mouse sepsis model. In pig sepsis, PPARα decline has been shown to precede hepatic steatosis, mirroring findings in CLP mice (Vandewalle et al, 2022). To enhance the applicability of our findings, we are conducting two clinical studies to collect liver biopsies from dogs with spontaneous septic peritonitis and humans with peritoneal sepsis.

Hepatic HNF4α is downregulated in NAFLD (Xu et al, 2015, 2021) and in advanced cirrhosis (Florentino et al, 2020) at the expression level, transcribed from an alternative promoter (P2 promoter) (Argemi et al, 2019) in AH, and accumulates in the

cytoplasm due to reduced acetylation in advanced cirrhosis (Florentino et al, 2020). However, during sepsis, HNF4α LOF primarily manifests at the level of chromatin binding. Despite *Hnf4a* mRNA downregulation in sepsis, total protein levels and the percentage of HNF4α-positive nuclei remained unchanged. Reduced HNF4α binding has been reported only in liver cancer, where it was caused by mutations in the Zn-finger DNA binding domain (Taniguchi et al, 2018).

We overlapped the HNF4α ChIP-Seq data with ATAC-Seq, H3K4me3 and H3K27ac ChIP-Seq data from livers 8 h after CLP, and investigated these epigenetic marks in the absence of hepatic HNF4α (using public data from Hnf4a^Liver-i-KO mice (Hunter et al, 2022; Qu et al, 2021)). Based on these analyses, we propose that alterations in HNF4α chromatin binding induced by CLP dictate H3K27 acetylation, and to some extent chromatin accessibility, but not H3K4 methylation. Hnf4a^Liver-i-KO mice show significant chromatin remodeling. Nevertheless, chromatin accessibility at differential HNF4α binding sites is only modestly changed during sepsis. We suggest that this is due to manifestation of alterations in HNF4α binding mainly as variations in binding intensity, rather than as binding loss or acquisition upon sepsis. This observation also elucidates the significant reduction in acetylation at CLP-specific HNF4α binding sites in the absence of hepatic HNF4α. Our data and supporting literature indicate that HNF4α may play a role in shaping the epigenetic landscape and regulating transcription in sepsis (Hunter et al, 2022; Qu et al, 2021). This literature suggests that histone acetylases are either not recruited or unable to function at HNF4α enhancers in the absence of hepatic HNF4α, even with the presence of other TFs (Qu et al, 2021). However, some studies propose that H3K27ac alone may be insufficient to determine enhancer activity (Zhang et al, 2020). Consequently, we cannot definitely claim that the alterations in HNF4α chromatin binding in CLP directly influence transcription without additional analyses, such as ChIP-Seq assessments of other epigenetic modifications or PRO-Seq experiments. However, the active histone mark H3K4me1 is also affected by hepatic HNF4α depletion (Qu et al, 2021).

The precise upstream mechanism causing the changes in HNF4α chromatin binding during CLP remains uncertain. Additional analyses showed no difference in HNF4α isoform abundance between sham and CLP conditions and demonstrated a significant reduction in HNF4α binding in vitro following CLP, suggesting that during sepsis, HNF4α may be either modified or exhibit reduced interaction with coactivators. Phosphorylation mediated by AMPK and ERK1/2 kinase has been described to impact the DNA binding capacity of HNF4α (Vetö et al, 2017; Hong et al,

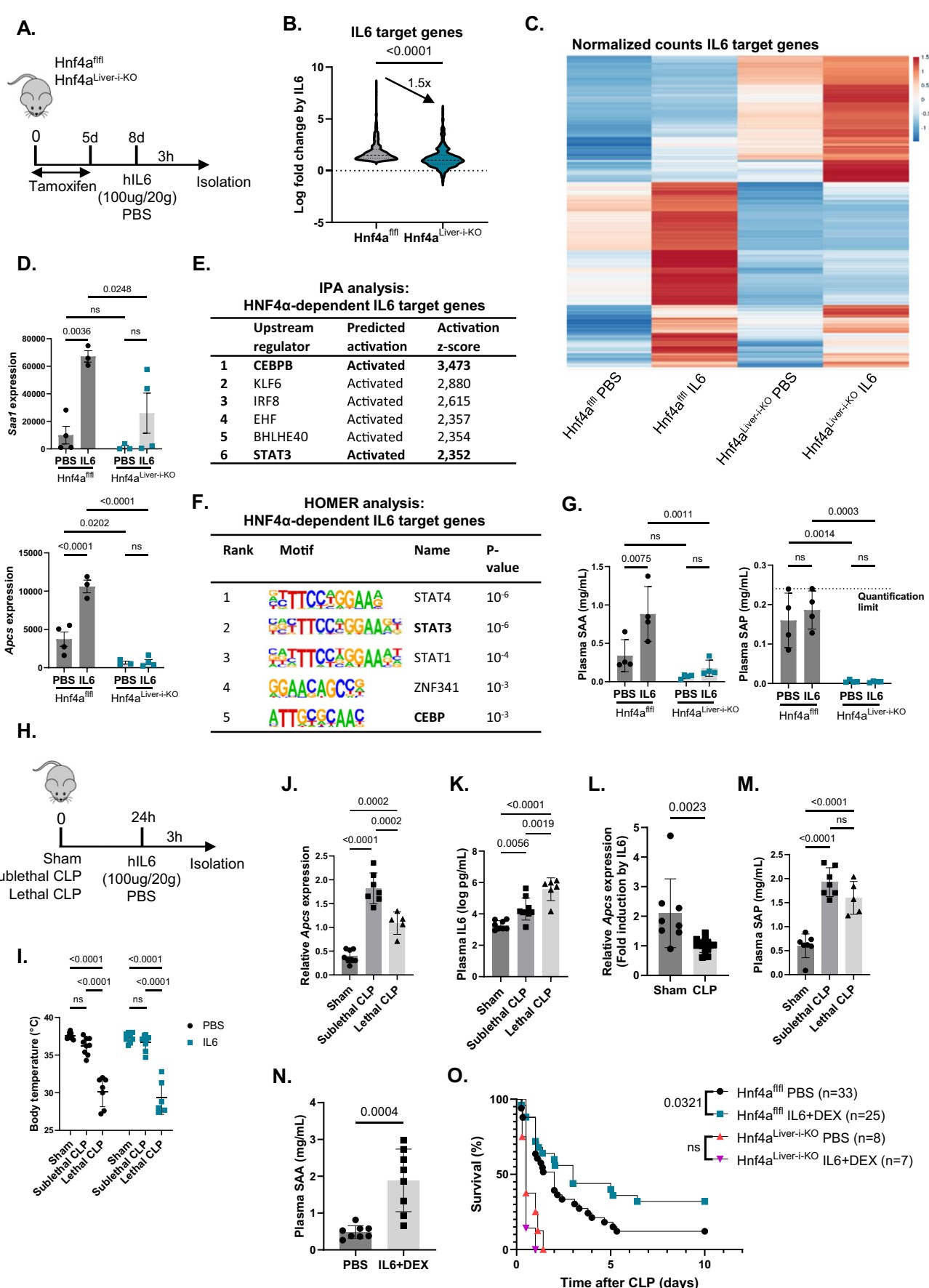

**Figure 6. Hepatic HNF4α loss-of-function results in reduced IL6-mediated C/EBPβ and STAT3 controlled acute phase response in lethal sepsis.**

(A–G) Hnf4a[Liver-i-KO] and Hnf4a[flfl] mice were treated with tamoxifen for 5 consecutive days, *i.p.* injected with hIL6 (100 μg/20 g) or PBS 3 days later, and liver (for RNA-Seq) and blood (for ELISA) were isolated 3 h later. n = 3–4/group (8 female, 7 male). IL6 target genes were defined by LFC > 0 and Padj < 0.05 in Hnf4a[flfl] IL6 relative to PBS. *P*-values were calculated from DESeq2 (Wald test). (A) Experimental setup. (B) Log fold change in IL6-induced expression of its target genes with LFC > 1 and Padj < 0.05 (n = 366/group). (C) Heatmap of IL6 target genes (unit scale bar: normalized counts). Rows are centered by unit variance scaling and clustered using correlation distance and average linkage. (D) Normalized counts of IL6 target genes *Saa1* and *Apcs*. (E, F) Analyses of IL6 target genes less induced by IL6 in Hnf4a[Liver-i-KO] relative to Hnf4a[flfl], and downregulated in Hnf4a[Liver-i-KO] PBS relative to Hnf4a[flfl] PBS (LFC < 0, Padj < 0.05). (E) IPA analysis of upstream regulators. The activation z-score predicts the activation state of the upstream regulator. (F) HOMER transcription factor motif enrichment 500 bp upstream of TSS. *P*-values derived from Fisher's exact test (Hypergeometric test). (G) Plasma serum amyloid A (SAA) and serum amyloid P (SAP) levels. n = 4/group (8 female, 8 male). (H–M) Mice were *i.p.* injected with hIL6 (100 μg/20 g) or PBS 24 h after sham or CLP. Liver (for qPCR) and blood (for ELISA) were isolated 3 h later. Sublethal *vs.* lethal CLP were distinguished by body temperature. n = 5–9/group. (H) Experimental setup. (I) Body temperature. (J) RT-qPCR *Apcs* mRNA expression relative to *Hprt* and *Rpl* in livers of PBS-injected mice. (K) Plasma IL6 levels in PBS-injected mice. (L) RT-qPCR *Apcs* mRNA expression in liver (fold induction by IL6) relative to *Hprt* and *Rpl*. (M) Plasma SAP levels in PBS-injected mice. (N, O) Mice were *i.p.* injected with hIL6 (5 mg/kg) + Dex (10 mg/kg) or PBS 12 h before CLP. (N) Plasma SAA levels at CLP onset. n = 8/group. (O) Mortality was monitored over 10 days. Data were analyzed with a one-sided chi-squared test. n = 73 (15 female, 58 male). Bars: mean ± SD, except for (E) mean ± SEM. Each dot represents a single biological replicate. *P*-values were analyzed with two-way ANOVA (D, G, I), one-way ANOVA (J, K, M) or unpaired t-test (B, L, N). ns: non-significant. Source data are available online for this figure.

2003). However, AMPK KO mice are more sensitive to sepsis, while AMPK activation offers protection (Jin et al, 2020; Kikuchi et al, 2020). Conversely, inhibition of ERK1/2 enhances survival in both the LPS and CLP models (Kopczynski et al, 2021). These findings suggest that by targeting upstream regulators like AMPK and ERK1/2, a far broader array of downstream targets will be impacted compared to specifically targeting HNF4α. Furthermore, the interaction between HNF4α and its coactivators is crucial for maintaining its active conformation and impact its DNA binding capacity (Duda et al, 2004). Most HNF4α coactivators show similar expression in sham and CLP, except for *Ppargc1a* (encoding PGC1α), which is significantly downregulated in CLP at 6 h and merits further investigation. While the PGC1α activator ZLN005 improves survival in polymicrobial sepsis, its impact on HNF4α is not yet understood (Suzuki et al, 2023). Moreover, different TF motifs were enriched in sham-specific and CLP-specific HNF4α ChIP-Seq and ATAC-Seq regions, suggesting a potential alteration in the HNF4α interactome triggered by sepsis. These interactors may dictate the function of HNF4α in sepsis, or vice versa.

To investigate the downstream effects of hepatic HNF4α LOF in sepsis, we generated Hnf4a[Liver-i-KO] mice. HNF4α depletion did not affect plasma FFA levels basally, which contrasts with another report describing increased lipolysis in Hnf4a[Liver-i-KO] mice (Huck et al, 2021). Our results support the notion that FFAs are transported from the blood into the liver, where they accumulate and form lipid droplets due to decreased fatty acid oxidation and VLDL secretion (Hayhurst et al, 2001; Yin et al, 2011; Martinez-Jimenez et al, 2010). During sepsis, FFAs are released into the blood after lipolysis of triglycerides in white adipose tissue and are absorbed by the liver, leading to accumulation and subsequent lipotoxicity, likely due to PPARα LOF (Van Wyngene et al, 2020). Due to the increased lethality of sepsis in Hnf4a[Liver-i-KO] mice, we expected HNF4α depletion to worsen hepatic steatosis. However, our findings did not align with this expectation, possibly because the liver had reached its maximum lipid storage capacity. Once this threshold is reached, excess FFAs may accumulate in the bloodstream and be absorbed by other organs such as the kidneys and heart, as we observed. Our work demonstrates that during homeostasis, HNF4α regulates lipid catabolism by (1) increasing the expression of PPARα, the primary TF responsible for inducing genes encoding lipid transporters, acyl-CoA synthetase and enzymes catalyzing β-oxidation and ketogenesis (Pineda Torra

et al, 2002; Chamouton and Latruffe, 2012) and (2) directly binding to promoters of several FFA oxidation genes, such as *Acsl5*, *Acsf2*, *Abcd1*, *Acox1*, and *Ehhadh* (Chen et al, 2020), and stimulating their expression. Our data further suggest that during sepsis, FFA metabolism is changed, mainly at the level of FFA oxidation and lipoprotein secretion, and that PPARα expression is downregulated, both of which can be attributed to the LOF of hepatic HNF4α. The decreased expression of PPARα in Hnf4a[Liver-KO] mice has been documented (Kasano-Camones et al, 2023), yet the relation between HNF4α and PPARα in sepsis has not yet been described.

Our data suggest that the loss of hepatic HNF4α impairs the expression of several other nuclear receptors besides PPARα, including RXRα, FXR, CAR, AR and LXRα, and therefore has a broad downstream impact in sepsis. The reduced expression of RXRα, FXR and LXRα in sepsis has already been described (Wang et al, 2011a; Hao et al, 2017; Chen et al, 2007). FXR regulates genes involved in bile acid metabolism and is crucial in cholestasis, a condition common in critically ill sepsis patients, associated with increased mortality and characterized by impaired bile secretion and bile acid accumulation in the circulation (Horvatits et al, 2017). Experimental cholestasis sensitizes mice to LPS-induced sepsis, while cholestyramine, a bile acid sequestrant, provides partial protection. FXR overexpression restores target gene expression, reduces serum bile acid levels, and partially protects against LPS-induced septic shock, whereas FXR KO mice are more sensitive (Hao et al, 2017). LXRα regulates genes involved in (lipid) metabolism, inflammation, and apoptosis. Septic mice or rats treated with the LXR agonist GW3965 show decreased multi-organ injury and reduced inflammatory cytokine levels, while LXRα-deficient mice have increased liver injury (Wang et al, 2011b; Zhang et al, 2021). Although CAR has not been studied in sepsis, many hepatic and intestinal cytochrome P450 enzymes and drug transporters are downregulated in sepsis (Lv and Huang, 2020). Targeting HNF4α could affect multiple nuclear receptors beyond the HNF4α-PPARα axis, potentially improving sepsis-associated hepatic steatosis, cholestasis, apoptosis, and alterations in drug metabolism.

Furthermore, we found that the CEBP motif was enriched within chromatin sites gaining HNF4α binding and/or accessibility in sepsis. C/EBPβ and STAT3 are known mediators of the hepatic acute phase response (APR). They are activated by IL6 as part of two signaling pathways: JAK-STAT and MAPK-CEBPβ

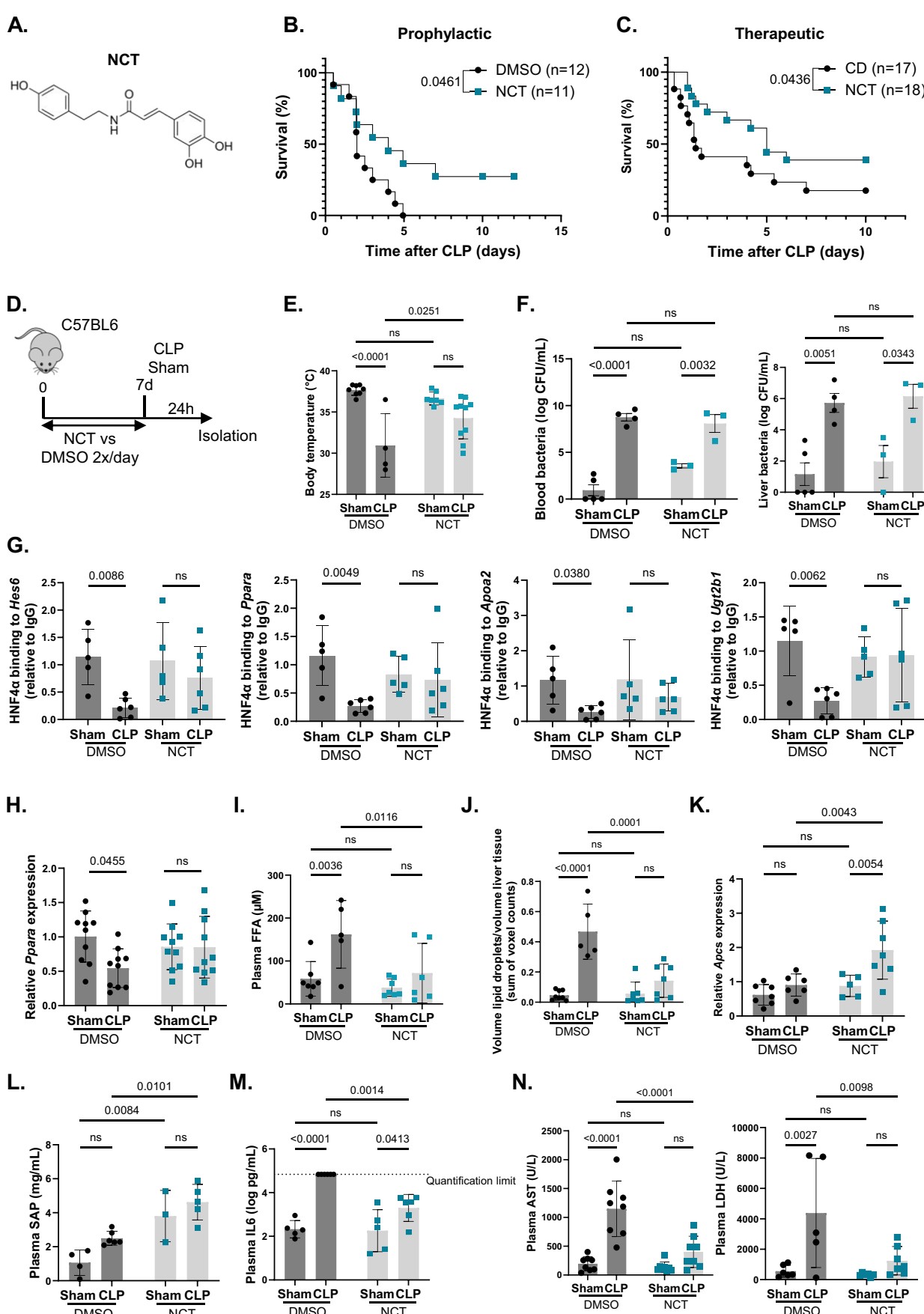

**Figure 7. HNF4α agonist NCT partially protects against polymicrobial sepsis by reducing hepatic steatosis and improving hepatic acute phase response.**

(A) Structural formula of N-trans-caffeoyltyramine (NCT). (B) Mice were injected with NCT (200 mg/kg) or DMSO *i.p.* twice daily for 6 days and subjected to CLP on day 7. Mortality was monitored over 12 days, during which NCT was injected. Data were analyzed with a Log-rank test. $n = 11$–12/group. (C) Mice were injected with NCT (200 mg/kg) or vehicle *i.p.* 3 h and 24 h post-CLP. Mortality was monitored over 10 days. Data were analyzed with a Gehan-Breslow-Wilcoxon test. $n = 17$–18/group. (D–N) Mice were injected with NCT (200 mg/kg) or DMSO *i.p.* twice daily for 6 days, subjected to sham or CLP on day 7, and liver and blood were isolated 24 h later. $n = 3$–8/group. (D) Experimental setup. (E) Body temperature. (F) Systemic and hepatic bacterial load (log CFU/mL). (G) HNF4α ChIP-qPCR relative to mouse IgG, on *Hes6, Ppara, Apoa2* and *Ugt2b1* promoters in liver. (H) RT-qPCR *Ppara* mRNA expression in liver relative to *Hprt* and *Rpl*. (I) Plasma FFA levels. (J) Lipid droplets volume (voxel counts) relative to liver tissue volume was calculated for each z-stack and averaged over all z-stacks per section. (K) RT-qPCR *Apcs* mRNA expression in liver relative to *Hprt* and *Rpl*. (L–N) Plasma serum amyloid P (SAP), IL6, aspartate aminotransferase (AST) and lactate dehydrogenase (LDH) levels. Bars: mean ± SD, except for (E, F) mean ± SEM. Each dot represents a single biological replicate. *P*-values were analyzed with two-way ANOVA. ns: non-significant. Source data are available online for this figure.

(Alonzi et al, 2001; Goldstein et al, 2017; Akira, 1997). Our data suggest that HNF4α is required for a proper IL6-mediated APR, which appears to be diminished during sepsis despite elevated plasma IL6 levels. Consequently, HNF4α LOF during sepsis not only leads to metabolic dysfunction but also impairs acute phase signaling. We further demonstrate that IL6 and DEX-mediated induction of the APR is partially protective in CLP. This was somewhat expected because several acute phase proteins (APPs) have protective activities in sepsis and sepsis-like models (Libert et al, 1994a; Hochepied et al, 2000; Van Molle et al, 1997; Dalli et al, 2014; Libert et al, 1996; Janz et al, 2013; Larsen et al, 2010). IL6, via STAT3 activation, also affects the degradation and synthesis of fatty acids in the liver, depending on the context (Gavito et al, 2016). Chronic high-dose IL6 (µg/g range), for example, reduced steatosis in 3 fatty liver disease models (Hong et al, 2004). However, a single dose did not affect FFA accumulation in the blood and liver in our CLP model, suggesting the protection is mainly due to increased acute phase proteins. Nonetheless, besides the anti-inflammatory and regenerative activities of IL6, it also acts as a pro-inflammatory cytokine, making its protective role in sepsis controversial (Libert et al, 1992; Libert et al, 1994b; Deutschman et al, 2006; Riedemann et al, 2003). Besides their recognized anti-inflammatory and/or antimicrobial functions, APPs serve diverse physiological roles. Notably, haptoglobin and hemopexin are pivotal in the clearance of hemoglobin and heme released during hemolysis (Smith and McCulloh, 2015; Larsen et al, 2010), while IL6 and APR stimulate hepatocellular regeneration (Moshage, 1997; Streetz et al, 2000; Cressman et al, 1996; Greenbaum et al, 1998). To conclude, our data suggest that HNF4α dysfunction in sepsis prevents an adequate APR towards IL6, potentially compromising liver regeneration and the removal of hemoglobin and heme during hemolysis. Whether HNF4α regulates IL6-mediated APR in sepsis directly or indirectly requires further study.

The suitability of HNF4α as a target for small drug discovery has been controversial (Yuan et al, 2009; Dhe-Paganon et al, 2002; Wisely et al, 2002). Some studies suggest that fatty acids serve as endogenous ligands of HNF4α, constitutively binding its ligand-binding domain (Wisely et al, 2002; Dhe-Paganon et al, 2002). However, we believe that the lack of effective, validated, synthetic HNF4α ligands is due not only to constitutive endogenous ligand binding but also to suboptimal screening strategies. The only compound proposed to specifically activate HNF4α in the presence of its endogenous ligand is NCT (Lee et al, 2021). However, it is uncertain whether NCT binds HNF4α directly in the ligand binding domain, binds an allosteric site, stabilizes the protein, or modulates its protein–protein interactions. Our data show that NCT reduces

sepsis lethality both prophylactically and therapeutically. We attribute this to decreased hepatic steatosis, inflammation and organ damage, and improved hepatic APR.

We conclude that peritoneal sepsis causes a rapid, progressive hepatic HNF4α LOF characterized by alterations in HNF4α chromatin binding, significantly affecting the epigenome and liver metabolic function. Specifically, hepatic HNF4α failure in sepsis mediates lipid dysfunction, downregulates several nuclear receptors, including PPARα, and diminishes IL6-mediated acute phase signaling. Most of these effects were reversed by administering the HNF4α agonist NCT. Based on our findings, we propose that HNF4α is a highly interesting target for translational studies and therapy in sepsis. However, more research is needed on the mechanism by which HNF4α chromatin binding is changed and the effect of HNF4α dysfunction on other liver metabolic pathways, such as gluconeogenesis and bile acid metabolism in sepsis. Our study focused on acute sepsis, and studies are needed on the relevance of our findings to chronic sepsis.

# Methods

**Reagents and tools table**

| Reagent/Resource | Reference or Source | Identifier or Catalog Number |
|---|---|---|
| **Experimental models** | | |
| C57BL/6J (*M. musculus*) | Janvier | N/A |
| Hnf4a^fl/fl^: B6.129×1(FVB)-Hnf4a^tm1.1Gonz^ (*M. musculus*) | Prof. Ioannis Talianidis (IMBB-FoRTH, Greece) | Strain# 004665, RRID:IMSR_JAX:004665 |
| AlbCreERT2^Tg/+^: SA^+/CreERT2^ (*M. musculus*) | Prof. Daniel Metzger and Pierre Chambon (Igbmc, France) | N/A |
| Alb-uPA^+/+^-SCID (*M. musculus*) | Prof. Philip Meuleman (UGhent, Belgium) | N/A |
| **Recombinant proteins** | | |
| hIL6 | VIB protein core | N/A |
| **Antibodies** | | |
| Human HNF4α antibody (Cl H1415) | R&D Systems | Cat# PP-H1415-0C |
| PPARα antibody | Abcam | Cat# ab126285, RRID:AB_3073567 |

| Reagent/Resource | Reference or Source | Identifier or Catalog Number |
|---|---|---|
| H3K27ac antibody | Active Motif | Cat# 39133, RRID:AB_2561016 |
| H3K4me3 antibody | Sigma-Aldrich | Cat# 07-473, RRID:AB_1977252 |
| Beta-actin antibody (BA3R) | Thermo Fisher Scientific | Cat# MA5-15739, RRID:AB_10979409 |
| Mouse IgG isotype control (ChIP-qPCR) | Thermo Fisher Scientific | Cat# 10400C, RRID:AB_2532980 |
| normal mouse IgG (ChIP-Seq) | Santa-Cruz Biotechnology | Cat# sc-2025 |
| Stat3 antibody (79D7) | Cell Signaling Technology | Cat# 4904, RRID:AB_331269 |
| Phospho-stat3 antibody (Tyr705) (D3A7) | Cell Signaling Technology | Cat# 9145 |
| Goat anti-Mouse IgG Secondary Antibody, Alexa Fluor™ 568 | Thermo Fisher Scientific | Cat# A-11031 |
| ECL™ Anti-Mouse IgG | GE Healthcare Life Sciences | Cat# GENA931 |
| ECL™ Anti-Rabbit IgG | GE Healthcare Life Sciences | Cat# GENA934 |
| **Oligonucleotides and other sequence-based reagents** | | |
| qPCR primers | This MS | Appendix Table S6 |
| ChIP-qPCR primers | This MS | Appendix Table S7 |
| **Chemicals, Enzymes and other reagents** | | |
| Rapidexon 2 mg/ml (DEX) | Dechra | N/A |
| Metronidazole | Sigma-Aldrich | Cat# M-1547 |
| Ceftriaxone | Sigma-Aldrich | Cat# C5793 |
| Tamoxifen | Sigma-Aldrich | Cat# T5648 |
| GW7647 | Tocris Bioscience | Cat# 1677 |
| NCT | SynBioC | N/A |
| LipidTox Deep Red | Thermo Fisher Scientific | Cat# H34477 |
| DAPI | Roche | Cat# 10236276001 |
| Acti-Stain 488 Phalloidin | Cytoskeleton | Cat# PHDG1-A |
| **Software** | | |
| GraphPad Prism v.8 | GraphPad Software | GraphPad Prism, RRID: SCR_002798 |
| HOMER | (Heinz et al, 2010) | HOMER, RRID:SCR_010881 |
| Metascape | (Zhou et al, 2019) | Metascape, RRID:SCR_016620 |
| Enrichr | (Xie et al, 2021), (Chen et al, 2013), (Kuleshov et al, 2016) | Enrichr, RRID:SCR_001575 |
| IPA | ingenuity pathway analysis (Qiagen) | IPA, RRID:SCR_008653 |
| ClustVis | (Metsalu and Vilo, 2015) | ClustVis, RRID:SCR_017133 |
| qbase | qbase+ software (Biogazelle, Gent, Belgium) | qBasePLUS, RRID:SCR_003370 |

| Reagent/Resource | Reference or Source | Identifier or Catalog Number |
|---|---|---|
| QuPath | (Bankhead et al, 2017) | QuPath, RRID:SCR_018257 |
| genrich | (Gaspar, 2018) | Genrich, RRID:SCR_002630 |
| BWA aligner | (Li and Durbin, 2009) | BWA, RRID:SCR_010910 |
| MACS2 | (Zhang et al, 2008) | MACS, RRID:SCR_013291 |
| DESeq2 | (Love et al, 2014) | DESeq2, RRID:SCR_015687 |
| **Other** | | |
| Aurum total RNA mini kit | Bio-Rad | Cat# 732-6820 |
| iScript cDNA synthesis kit | Bio-Rad | Cat# 170-8891 |
| SensiFast SYBR No-Rox Kit | GeC Biotech | Cat# CSA-01190 |
| WesternBright ECL kit | Advansta | Cat# K-12045-D20 |
| Free Fatty Acid Assay kit | Abnova | Cat# KA1667 |
| IL-6 mouse ELISA kit | Thermo Fisher Scientific | Cat# BMS603-2 |
| Mouse SAA ELISA Kit | Immunology Consultants Laboratory | Cat# E-90SAA |
| Mouse SAP ELISA Kit | Immunology Consultants Laboratory | Cat# E-90SAP |
| Protein G Sepharose 4 Fast Flow (ChIP-qPCR) | Cytiva | Cat# GE17-0618-01 |
| Dynabeads™ protein G (ChIP-Seq) | Thermo Fisher Scientific | Cat# 10003D |
| QIAquick PCR Purification Kit (ChIP-qPCR) | Qiagen | Cat# 28106 |
| Monarch PCR&DNA Cleanup kit (ChIP-Seq) | Biolabs | Cat# T1030 |
| Qubit 1x dsDNA high sensitivity kit | Thermo Fisher Scientific | Cat# Q33230 |
| NEBNext Ultra II DNA Library Prep kit | Biolabs | Cat# E7645 |
| Hepatocyte Nuclear Factor 4 Alpha/Gamma (HNF4A/HNF4G) DNA-Binding ELISA Kit | Abbexa | Cat# abx596563 |

## Methods and protocols

### Mice

Male C57BL/6J mice were ordered from Janvier (Le Genest-St. Isle, France). Mutant Hnf4a$^{fl/fl}$ mice were generated by Dr. Frank Gonzalez (NIH, Bethesda, USA) and formally called B6.129×1(FVB)-*Hnf4a$^{tm1.1Gonz}$*/J (Hayhurst et al, 2001). Exons 4 and 5 were flanked by loxP sites using ES cell technology. The mice had been backcrossed into C57BL/6J background and were provided by Dr. Iannis Talianidis (University of Crete, Heraklion, Greece), by courtesy of Dr. Frank Gonzalez, and were under protection of an MTA. Subsequently, we crossed these mice with AlbCreERT2$^{Tg/+}$ mice, which were kindly provided by Dr. D. Metzger & Dr. P. Chambon (Igbmc, France) (Schuler et al, 2004).

The humanized liver mice were generated by Prof. Philip Meuleman (UGhent, Belgium) as described in (Meuleman et al, 2005). Briefly, the mice were generated by transplanting homozygous Alb-uPA$^{+/+}$-SCID mice (which suffer from spontaneous and chronic death of hepatocytes) with $0.7 \times 10^6$ cryopreserved primary human hepatocytes (donor C342, Lonza), via intrasplenic injection. The human albumin concentration in plasma was determined 6 weeks after transplantation by ELISA (Bethyl Laboratories, Montgomery, Texas, United States) and was used as a marker of liver chimerism.

All mice were housed in individually ventilated cages (IVC) at standard housing conditions (22 °C, 14/10 h light/dark cycle) with food (chow diet consisting of 18% proteins, 4.5% fibers, 4.5% fat, 6.3% ashes, Provimi Kliba SA) and water ad libitum (except when otherwise stated: ChIP-Seq & ATAC-Seq experiments). The IVC were kept in a specific pathogen free facility. Both male and female offspring was used between 8 and 20 weeks. All experiments were approved by the institutional ethics committee for animal welfare of the Faculty of Sciences, Ghent University, Belgium (EC2020-091, EC2023-019, EC2024-059). The methods were carried out in accordance with the relevant guidelines and regulations.

Mice were subjected to CLP operation (Rittirsch et al, 2009). Briefly, mice were anesthetized with isoflurane, followed by a 1 cm incision in the abdominal skin. The cecum was ligated for 75% and punctured twice with a 21-gauche needle, which allows the cecal content to leak into the abdomen and cause a systemic infection. The abdominal musculature and skin were closed with simple running sutures and metallic clips, respectively. Sham mice underwent the same procedure without ligating and puncturing the cecum. During lethality studies, mice were injected intraperitoneally (IP) with broad-spectrum antibiotics (25 mg/kg ceftriaxone and 12.5 mg/kg metronidazole) in phosphate buffered saline (PBS) 8 h and 24 h after CLP. Rectal BT was measured 3 times a day and mice were euthanized when their BT dropped below 28 °C (=humane endpoint).

To induce hepatic HNF4α depletion, Hnf4a$^{fl/fl}$;AlbCreERT2$^{Tg/+}$ and Hnf4a$^{fl/fl}$;AlbCreERT2$^{+/+}$ mice received IP injections of 1 mg tamoxifen in a 1:8 ethanol:oil solution for 5 consecutive days. Complete depletion of HNF4α was observed 3 days after the final tamoxifen injection. To investigate the IL6-mediated response in the absence of hepatic HNF4α and during sepsis, Hnf4a$^{Liver-i-KO}$ mice or mice subjected to CLP were IP injected with human IL6 (hIL6) (5 mg/kg, dissolved in PBS; synthetized by VIB protein core and free of endotoxin contamination), either 3 days after the last tamoxifen injection or 24 h after sham or CLP surgery, respectively. To study the APR-mediated protection in sepsis, mice received IP injection of a combination of hIL6 (5 mg/kg) and dexamethasone (DEX) (10 mg/kg) or PBS, 12 before CLP induction. Additionally, mice were subjected to an 8 h fasting period following the CLP procedure for ChIP-Seq and ATAC-Seq experiments. To assess PPARα activity in the liver after HNF4α depletion, Hnf4a$^{Liver-i-KO}$ mice were IP injected with GW7647 (7.5 mg/kg, dissolved in DMSO; Tocris Bioscience), 3 days after the final tamoxifen injection. Additionally, the HNF4α agonist NCT was administered IP twice daily for 6 days prior to CLP induction at a dose of 200 mg/kg, dissolved in DMSO, with continued administration during the CLP operation. For prophylactic lethality studies, mice were also treated with NCT following CLP induction. Due to the known toxicity of DMSO, 2-hydroxypropyl-β-cyclodextrin (CD) was utilized as a vehicle for NCT administration in a therapeutic context. Mice received two doses of NCT, one administered 3 h post-CLP and another 24 h post-CLP. NCT was synthetized by SynBioC (Dr. Bart Roman).

### Liver transcriptomic analysis

Liver was isolated and stored in RNAlater (Life Technologies Europe), after which RNA was obtained with the Aurum Total RNA Mini kit (Bio-Rad) according to the manufacturer's-protocol. RNA concentration was measured and quality was determined by the Nanodrop 1000 (Thermo Scientific).

Real-time qPCR.   1000 ng RNA was reverse-transcribed into cDNA using the iScript cDNA Synthesis kit (Roche). cDNA was diluted 10 times in nuclease-free water prior qPCR analysis, performed with the Light Cycler 480 system (Roche) using Sensifast Bioline Mix (Bio-Line). Samples were loaden in duplicate and housekeeping genes *Gapdh*, *Hprt* and/or *Rpl* were included to normalize target gene expression. Used qPCR primers and associated sequences are listed in Appendix Table S6. qPCR data was analyzed with qbase+ software (Biogazelle, Gent, Belgium).

RNA sequencing (RNA-Seq).   RNA concentration was measured with Nanodrop 8000 (Thermo Scientific), and RNA quality was checked with 5300 Fragment Analyzer (Agilent). RNA was used to generate an Illumina sequencing library using the Illumina Stranded mRNA Ligation LP kit (IL6 HNF4αKO RNA-Seq) or Illumina TruSeq Stranded mRNA kit (HNF4αKO CLP RNA-Seq), and single-end sequencing was done with Illumina NovaSeq 6000 (VIB nucleomics core). Single-end 100 bp reads were mapped to the mouse (mm10) reference genome with STAR (Dobin et al, 2013). Multimapping reads were further excluded from the data processing. Differential expression analysis was done using the DESeq2 package (Love et al, 2014), with the FDR set at 5%. RNA-Seq data was analyzed with various tools: Enrichr (Xie et al, 2021; Chen et al, 2013; Kuleshov et al, 2016), HOMER (Heinz et al, 2010), Metascape (Zhou et al, 2019), ClustVis (Metsalu and Vilo, 2015) and ingenuity pathway analysis (IPA) (Qiagen). Processing of CLP 8 h & 24 h RNA-Seq is described in (Vandewalle et al, 2021). Identity of the genes related to Fig. 1B and 1D can be found in Dataset EV1.

### Western blot analysis

Total protein was isolated from snap-frozen livers with RIPA lysis buffer, supplemented with protease inhibitor cocktail (Roche). Protein concentration was determined by Bradford assay. Protein samples containing 30 μg of protein and loading dye were separated by electrophoresis on a 8% SDS-polyacrylamide gel and transferred to a nitrocellulose membrane (with a pore size of 0.45 μm) by blotting. The membrane was blocked with a ½ dilution of Starting Block/PBST0,1% (Thermo Fisher Scientific) and incubated overnight at 4 °C with primary antibodies against HNF4α (1:1000; PP-H1415-0C, R&D Systems), PPARα (1:1000; ab126285, Abcam), STAT3 (1:2000; 4904, Cell Signaling Technology), phospho-STAT3 (1:2000; 9145, Cell Signaling Technology) and β-actin (1:5000; BA3R, Thermo Fisher Scientific), as an internal control. After washing with PBST0,1%, Amersham ECL anti-mouse antibody (1:2000; GENA931, GE Healthcare Life Sciences) or Amersham ECL anti-rabbit antibody (1:2000; GEN934, GE Healthcare Life Sciences) was added to the membrane for 1 h at room temperature (RT). The membrane was again washed with PBST0, 1% and

immunoreactive bands were visualized and quantified using Amersham Imager 600 (GE Healthcare Life Sciences) and the WesternBright Quantum kit (advansta).

### Histological analysis

Immunohistochemistry.   Liver was isolated and fixed overnight at 4 °C in 4% PFA. After transferring to 70% ethanol, livers were dehydrated within the Shandon Citadel 2000 (Thermo Fisher Scientific) and bedded in paraffin. Paraffin slices of 5 μm in thickness were dewaxed and rehydrated using the ST5010 Auto-stainer XL (Leica). After antigen retrieval inside a PickCell pressure cooker, liver slices were washed with PBS and blocked with PBS containing 0.5% BSA, 0.1% Tween-20 and 1% goat serum. Paraffin sections were incubated overnight at 4 °C with primary antibody against HNF4α (1:100; PP-H1415-00, R&D Systems) or blocking buffer (=negative control samples). After being washed in PBS, the slices were incubating with a goat anti-mouse antibody coupled to AF568 (1:500; A11031, Thermo Fisher Scientific). The slices were washed again and DAPI nuclear staining (1:1000; Roche) was added for 15 min at RT. Finally, the liver sections were rinsed with PBS, mounted with PVA including DABCO and imaged with Axioscan.Z1 Slide scanner (Zeiss), using a Plan-Apochromat 10x/0.45 M27 objective lens at a pixel size of 0.650 μm. Quantification of percentage HNF4α-positive nuclei was done in QuPath (Bankhead et al, 2017).

LipidTox.   Liver was isolated and fixed in antigenfix (DiaPath) at 4 °C for 1–2 h. After being washed in PBS, liver pieces were put overnight in 34% sucrose at 4 °C and were mounted with O.C.T. compound (Tissue-Tek). Cryosections of 20 μm in thickness were rehydrated in PBS and blocked in blocking buffer (2% BSA, 1% fetal calf serum, 1% goat serum, in 0.5% saponin) for 30 min at RT. The antibody mix (LipidTox Deep Red (1:400; Life Technologies Europe B.V.), Acti-stain 488 Phalloidin (1:150; Cytoskeleton Inc.)) was added and incubated for 2 h at RT. Negative control samples were only incubated with the Acti-stain. After washing the slides in PBS, DAPI nuclear staining (1:1000; Roche) was added for 15 min at RT. Finally, the liver sections were washed in PBS and Bidi to remove residual salts, mounted with PVA including DABCO and imaged with a spinning disk confocal microscope (Zeiss), using a Plan-Apochromat 40x/1.4 Oil DIC (UV) VIS-IR M27 objective lens at a pixel size of 0.275 μm and at optimal Z-resolution (240 mm) or a Plan-Apochromat 20x/0.8 M27 objective lens at a pixel size of 0.550 μm and at optimal Z-resolution (490 mm). Z-stacks of 8–10 areas/sample were imaged. The total volume of lipid droplets (depicted as voxel count) relative to the tissue volume was calculated in Volocity (PerkinElmer).

### Chromatin immunoprecipitation (ChIP)

ChIP was performed on 200–400 mg of snap-frozen liver tissue. First, Dynabeads™ protein G (Thermo Fisher Scientific) were washed three times with BSA/PBS and coupled to a primary antibody against HNF4α (3 μg/IP sample; PP-H1415-00, R&D Systems), H3K27ac (3 μg/IP sample; 39133, Active Motif), H3K4me3 (3 μg/IP sample; 07-473, Sigma-Aldrich) or mouse IgG (3 μg/IP sample; sc-2025, SantaCruz), as an internal control, overnight at 4 °C. Afterwards, the beads with coupled antibody were washed with BSA/PBS twice. Livers were homogenized and fixed in Farnham Lysis buffer with 1% formaldehyde, supplemented

with protease inhibitor cocktail (Roche), for 10 min at RT. Released nuclei were isolated by centrifugation at 2000 rpm for 5 min and washed with PBS, supplemented with protease inhibitor cocktail (Roche). The nuclei were further lysed in RIPA buffer, supplemented with protease inhibitor cocktail (Roche), and sonicated (Bioruptor, Diagenode) at 4 °C for 40 cycles (30 s ON, 30 s OFF) with high settings to yield 200–800 bp DNA fragments. Cell debris was removed by centrifugation at full speed for 20-30 min at 4 °C. Sonication efficiency was checked (after reverse crosslinking with proteinase K for 1 h at 65 °C and DNA purification) on a 1.5% agarose gel. For immunoprecipitation (IP), antibody-coupled magnetic beads were added to the sonicated samples and incubated overnight at 4 °C. Input samples containing sonicated lysate were used as a control during analysis. The day after, bead-antibody-DNA complexes were washed 5 times with LiCl wash buffer (100 mM Tris-HCl; pH 7.5, 500 mM LiCl, 1% IGEPAL, 1% sodium deoxycholate) at 4 °C and 2 times with TE buffer (10 mM Tris-HCl; pH 7.5, 0.1 mM EDTA) at 4 °C. Chromatin was eluted by incubating the IP samples with elution buffer (1% SDS, 0.1 M NaHCO₃) 1 h at 65 °C. Both IP and input samples were incubated with proteinase K for 4 h at 65 °C for reverse crosslinking and DNA was purified in 50 μL elution buffer according to the manufacturer's protocol by using the Monarch PCR&DNA Cleanup kit (Biolabs). DNA concentration was measured with Qubit Fluorometer using the Qubit 1x dsDNA high sensitivity kit (Thermo Fisher Scientific).

ChIP-qPCR.   ChIP-qPCR was performed on 2 μL of purified chromatin from IP and input samples with Light Cycler 480 system (Roche) using Sensifast Bioline Mix (Bio-Line), to measure HNF4α binding to HNF4α target gene promoters. qPCRs were carried out in duplicate. Used qPCR primers and associated sequences are listed in Appendix Table S7. Results after immunoprecipitation were subtracted from the input and expressed as relative enrichment to the negative IgG control.

ChIP-Seq.   From the purified chromatin, 15 ng was used to generate a ChIP-Seq library according to the manufacturer's protocol using NEBNext Ultra II DNA Library Prep kit (E7645L, NEB). The sample quality was checked with the 2100 Bioanalyzer using DNA 1000 kit (Agilent). Libraries were single-end sequenced by Illumina NextSeq 500 (75SE). The reads were mapped to the mouse (mm10) reference genome with BWA aligner (Li and Durbin, 2009). Multimapping reads were further excluded from the data processing. Peaks were called with MACS2 (Zhang et al, 2008) relative to the input sample, using the preset settings from the 'factor' internal method and requiring at minimum a 4-fold enrichment above the background and the control input sample. No spike-in controls were used for quantitative assessment, but peaks called in a certain condition had to be present in all 4 replicates (determined by direct region overlap). Differential peaks were identified by DESeq2 (Love et al, 2014), with the FDR set at 5%. HNF4α ChIP-Seq data was analyzed with HOMER motif enrichment (Heinz et al, 2010) (also to verify HNF4α binding) and metascape pathway analysis (Zhou et al, 2019). Identity of the genes related to Fig. 2 can be found in Dataset EV1.

### HNF4α DNA binding

The DNA binding activity of HNF4α in liver nuclear lysates was evaluated using ELISA (Abbexa) according to the manufacturer's

**The paper explained**

A critical role for HNF4α in polymicrobial sepsis-associated metabolic reprogramming and death.

**Problem**

Sepsis is among the most lethal medical conditions globally, affecting 49 million people annually and causing 11 million deaths, constituting nearly 20% of all global deaths. Inflammation was previously thought to be the primary cause of death, but recent evidence suggests that metabolic dysregulation also plays a significant role. Previous research in our lab has shown that sepsis induces a starvation response—aimed at preserving the energy levels—by releasing free fatty acids from white adipose tissue into the blood. These are subsequently taken up by liver epithelial cells (hepatocytes) where they should be oxidized to release energy or to be converted into ketone bodies. During sepsis, however, the hepatic PPARα expression, which transcriptionally controls these processes, is reduced, leading to the accumulation of free fatty acids in the blood and liver, causing toxicity. The mechanism by which PPARα is downregulated in sepsis remains unknown.

**Results**

In the current paper, we show that HNF4α loses its function in liver during sepsis, and that this loss-of-function is responsible for the PPARα-mediated problems we observed previously, as well as many others. We utilize the cecal ligation and puncture mouse model to study polymicrobial sepsis, which affects about 25% of all sepsis patients. The HNF4α loss-of-function is associated with reduced DNA binding, affecting the acetylation and to a lesser extent the accessibility status of these HNF4α binding sites and thereby the transcription of its target genes. We show that genetic depletion of hepatic HNF4α enhances sepsis lethality, confirming its critical role in this disease, associated with increased lipid dysfunction, inflammation, and cell death. In these HNF4α knockout mice, PPARα expression and activity are low, and blood free fatty acid levels and hepatic lipid content are further increased in the context of sepsis. In addition to the PPARα-mediated lipid problems, HNF4α loss-of-function also impairs the liver's ability to mount an adequate acute phase response to IL6, crucial for proper liver regeneration. Pre-administration of IL6 and dexamethasone, which maximally activate the acute phase response, protects against sepsis. Therefore, we believe that both the lipid-associated problems and the reduced acute phase response downstream of HNF4α loss-of-function contribute to lethality in sepsis. Furthermore, HNF4α depletion reduces not only the expression of PPARα but also other metabolic transcription factors such as CAR, RXRα, FXR and LXRα. Importantly, the HNF4α agonist NCT provides protection in sepsis across all these mentioned levels. Finally, we confirm hepatic HNF4α loss-of-function in septic pigs and a humanized liver mouse sepsis model, enhancing the translatability of the current findings.

**Impact**

In this study, we identify hepatic HNF4α as a critical mediator essential for maintaining liver metabolism, as well as acute phase response and liver regeneration, in healthy animals. We propose HNF4α as a promising therapeutic target in sepsis, highlighting the therapeutic potential of the HNF4α agonist NCT for both treatment and prevention. Furthermore, we emphasize PGC1α, a recognized upstream regulator of HNF4α, as another viable therapeutic target. Our findings are expected to significantly impact both the HNF4α field, particularly its role in hepatic acute phase response, and the sepsis field, emphasizing HNF4α as a potential therapeutic target and the HNF4α agonist NCT as a new treatment option.

instructions. Nuclear lysates were prepared from snap-frozen liver samples. First, the liver was homogenized in PBS supplemented with a protease inhibitor cocktail (Roche). Second, the nuclei were isolated using a lysis buffer containing 20% NP-40 and 10% Triton-X100, supplemented with a protease inhibitor cocktail (Roche).

Third, the nuclei were washed and lysed in a buffer containing 10% N-lauroylsarcosine and 10% Na-Deoxycholate, supplemented with a protease inhibitor cocktail (Roche). Finally, the samples were sonicated using a PIXUL sonicator (Active Motif).

### ATAC-Seq

Nuclei were isolated from snap-frozen liver tissue as described (10x genomics). Afterwards, nuclei isolation was verified and the nuclei were counted using 0.4% Trypan Blue (Sigma) and EVOS cell imaging (Thermo Fisher Scientific). Around 100 K nuclei were mixed with Tn5 Transposase loaded with sequencing adapters (Diagenode) and incubated for 45 min at 37 °C. Transposed DNA was purified in 11 μL elution buffer according to the manufacturer's protocol by using the Monarch PCR&DNA Cleanup kit (Biolabs). Afterwards, transposed DNA was amplified according to published protocol and PCR primers (Buenrostro et al, 2015). Next, PCR-amplified DNA fragments between 150 bp and 800 bp were selected from the library by using SPRIselect beads (Beckman coulter) according to manufacturer's instructions. The library quality was checked with 2100 Bioanalyzer (Agilent) using the high sensitivity DNA kit (Agilent). Samples from 4 biological replicates were paired-end sequenced with Illumina NextSeq 500 (40PE). The reads were mapped to the mouse (mm10) reference genome with BWA aligner (Li and Durbin, 2009). Multimappers and mitochondrial reads were further excluded from the data processing. Peaks were called with genrich (Gaspar, 2018), using the default settings in ATAC-seq mode, but with removal of PCR duplicates. Peaks called in a certain condition had to be present in all 4 replicates (determined by direct region overlap). Differential peaks were identified by direct region overlap and DESeq2 (Love et al, 2014), with the FDR set at 5%. ATAC-Seq data was analyzed with HOMER motif enrichment (Heinz et al, 2010) and metascape pathway analysis (Zhou et al, 2019). Identity of the genes related to Fig. 3 can be found in Dataset EV1.

### Biochemical analysis

Analysis of organ damage markers AST, LDH, and ALT in plasma was performed in the University Hospital of Ghent. Plasma FFA levels were determined via a colorimetric assay kit (Abnova). Plasma IL6 (eBioscience), SAA (ICL lab) and SAP (ICL lab) levels were measured by ELISA. All assays were performed according to the manufacturer's instructions.

### Bacterial load quantification

Bacterial load in the liver was determined by homogenizing the right part of the medial lobe in 1 mL sterile PBS. From this liver homogenate, 40 μL was plated out on half of tryptic soy agar (TSA) plates. On the other half, 40 μL of 1:2 diluted blood in sterile PBS was plated out to determine systemic bacterial load. Colony-forming units (CFUs) were counted 24 h post-inoculation at 37 °C and CFU/mL was calculated.

### Statistical analysis

Bacterial load, expressed as CFU/mL, and plasma IL6 levels were log-transformed. Also, qPCR data were log-transformed to obtain normal distribution. Normal distribution was checked using a normality test in Prism 9.0 (GraphPad Software, Inc), which was mainly used to plot the data. When comparing two group means, an unpaired student's t-test was used. If the standard deviation (SD) between the groups differed significantly, Welch's correction was implemented. For more than 2

groups, *P*-values were analyzed with a one-way ANOVA including Dunnett's or Tukey's multiple comparisons test. Two-way ANOVA including Tukey's or Sidak's multiple comparisons test was used for analyses with a second variable. Kaplan–Meier survival curves were compared using Log rank test, Gehan-Breslow-Wilcoxon test or one-sided chi-squared test. *P*-values of <0.05 were considered to be statistically significant. ns: not significant. n represents the number of biological replicates used for generating the data, which was determined on the basis of previous experiments. Each experiment shown in the main figures was repeated at least twice. Data are represented as dot plots with bars, showing the mean and SD or SEM. Statistical details can be found in the Figs and/or Fig legends. When allocating animals to a treatment, an equal number of animals of each sex and similar age were assigned to each treatment group to mitigate the influence of sex- or age-related factors. Each treatment was also allocated to different cages. Data analysis was done blindly to avoid subjective bias.

### Graphics

Synopsis image was created with BioRender.com.

## For more information

Author website: https://www.irc.ugent.be/index.php/groups/libert-unit.

## Data availability

Sequencing data have been deposited at the National Center for Biotechnology Information Gene Expression Omnibus (GEO) public database (http://www.ncbi.nlm.nih.gov/geo/) and are publicly available as of the date of publication under accession numbers: HNF4α ChIP-Seq data: GEO GSE245682. ATAC-Seq data: GEO GSE244821. H3K4me3 & H3K27ac ChIP-Seq data: GEO GSE260577. HNF4aKO RNA-Seq data: GEO GSE260635. IL6 HNF4aKO RNA-Seq data: GEO GSE246053. Microscopy images of the HNF4α immunohistochemistry have been deposited at Bioimage Archive (https://www.ebi.ac.uk/biostudies/bioimages/studies/) and are publicly available as of the date of publication under accession number S-BIAD1239.

The source data of this paper are collected in the following database record: biostudies:S-SCDT-10_1038-S44321-024-00130-1.

## Peer review information

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

## Acknowledgements

The authors would like to thank Joke Van den Berghe and the staff of the animal care facilities for animal care, as well as the staff of the VIB microscopy core for help with the visualization and quantification of the IHC and LipidTox samples. We acknowledge the VIB Nucleomics Core for sequencing analysis, the VIB Protein Core for hIL6 and SynBioc for NCT. We wish to thank Prof. Ioannis Talianidis (IMBB-FoRTH, Greece) for sharing Hnf4a[fl/fl] mice, as well as Prof. Daniel Metzger and Pierre Chambon (Igbmc, France) for providing AlbCre-ERT2 mice. We thank the lab for technical support, in particular Hester Dufoor, Elise Moens, Melanie Eggermont, Louise Nuyttens and Maxime Roes. This work was supported by an FWO (Fonds Wetenschappelijk Onderzoek) grant to C.V.D (11M6324N). Further funding for this project was obtained via the Research Foundation Flanders (FWO-Vlaanderen) (SBO STOP SEPSIS – S003122N), the Research Council of Ghent University (GOA and Methusalem Program).

## Author contributions

**Céline Van Dender**: Conceptualization; Formal analysis; Investigation; Visualization; Writing—original draft; Writing—review and editing. **Steven Timmermans**: Data curation; Software; Formal analysis. **Ville Paakinaho**: Resources; Writing—review and editing. **Tineke Vanderhaeghen**: Supervision; Writing—review and editing. **Jolien Vandewalle**: Supervision; Writing—review and editing. **Maarten Claes**: Resources. **Bruno Garcia**: Resources. **Bart Roman**: Resources. **Jan De Waele**: Supervision. **Siska Croubels**: Supervision. **Karolien De Bosscher**: Supervision; Writing—review and editing. **Philip Meuleman**: Resources. **Antoine Herpain**: Resources. **Jorma J Palvimo**: Resources. **Claude Libert**: Conceptualization; Resources; Supervision; Funding acquisition; Project administration; Writing—review and editing.

Source data underlying figure panels in this paper may have individual authorship assigned. Where available, figure panel/source data authorship is listed in the following database record: biostudies:S-SCDT-10_1038-S44321-024-00130-1.

## Disclosure and competing interests statement

The authors declare no competing interests. Claude Libert is a member of the Advisory Editorial Board of EMBO Molecular Medicine. This has no bearing on the editorial consideration of this article for publication.

# Expanded View Figures

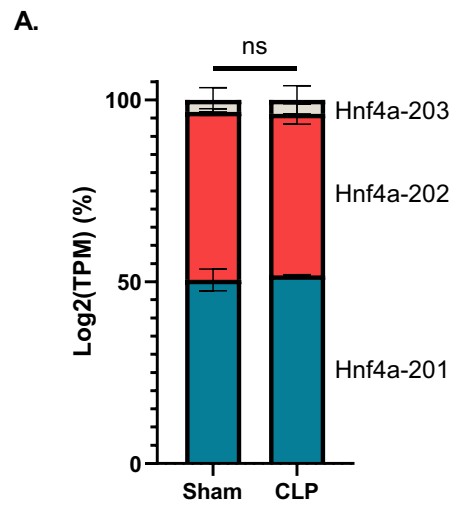

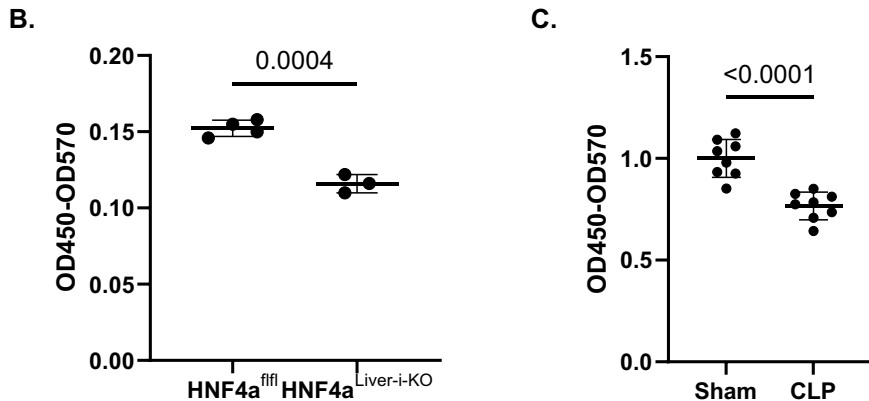

**Figure EV1. Absence of alternative splicing with reduced HNF4α DNA binding in septic liver.**

(A) Relative abundance of *Hnf4a* transcripts (Hnf4a-201, Hnf4a-202, and Hnf4a-203) in CLP relative to sham, expressed as log2(TPM) percentages, derived from in-house paired-end RNA-Seq data 8 h post-CLP. *n* = 3/group (biological replicates). (B, C) OD450-OD570 represents the strength of HNF4α binding to dsDNA oligos immobilized on a plate using nuclear lysates from tamoxifen-injected Hnf4a^flfl (*n* = 4) and Hnf4a^Liver-i-KO (*n* = 3) mice (B) or sham (*n* = 8) and CLP (*n* = 8) mice 8 h post-CLP (C). Bars: mean ± SEM (A), central lines: mean ± SD (B, C). Each dot represents a single biological replicate. *P*-values were analyzed with two-way ANOVA (A) or unpaired t-test (B, C). ns: nonsignificant.

**A.**

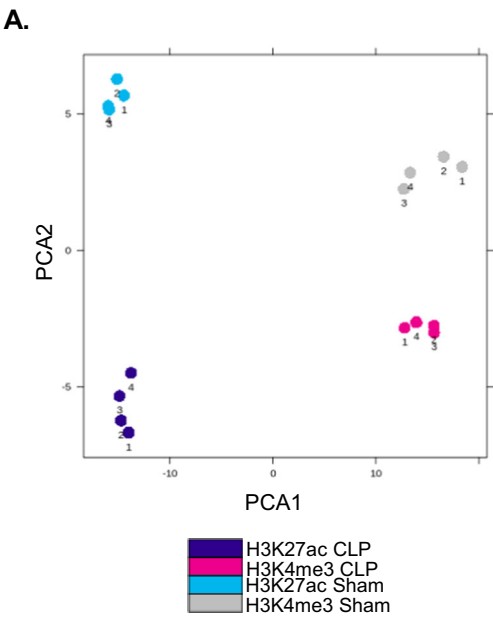

**B.**

**HOMER analysis: sham-specific open chromatin sites**

| Rank | Motif | Name | P-value |
|---|---|---|---|
| 1 | | HNF4a | $10^{-762}$ |
| 2 | | PPARa | $10^{-657}$ |
| 3 | | COUP-TFII | $10^{-493}$ |
| 4 | | RXR | $10^{-452}$ |
| 5 | | PPARE | $10^{-446}$ |

**HOMER analysis: CLP-specific open chromatin sites**

| Rank | Motif | Name | P-value |
|---|---|---|---|
| 1 | | CEBP | $10^{-338}$ |
| 2 | | CEBP:AP1 | $10^{-302}$ |
| 3 | | NFIL3 | $10^{-279}$ |
| 4 | | HLF | $10^{-234}$ |
| 5 | | Fos | $10^{-197}$ |

**C.**

◀ **Figure EV2. Perturbations in the chromatin binding dynamics of HNF4α during sepsis induce changes in the epigenetic landscape in the liver.**

(A–C) Liver was isolated 8 h after sham or CLP for ATAC-Seq, and H3K4me3 and H3K27ac ChIP-Seq analyses. $n = 4$/group (biological replicates). (A) H3K4me3 and H3K27ac ChIP-Seq PCA plot. (B) HOMER transcription factor motif enrichment 200 bp centered on the center of all differential peaks. *P*-values derived from Fisher's exact test (Hypergeometric test). Differential peaks were identified by direct region overlap and DESeq2 (using Wald test), with the FDR set at 5%. (C) Heatmaps representing H3K27ac and H3K4me3 signals from sham and CLP or Hnf4a^flfl and Hnf4a^Liver-i-KO 1.5 kbp or 1.0 kbp, respectively, centered on the center of differential HNF4α ChIP-Seq peaks with decreased intensity ('sham-specific') and increased intensity ('CLP-specific') in CLP. Differential peaks were identified by DESeq2 (using Wald test), with the FDR set at 5%.

## HNF4α-dependent pathways affected upon CLP

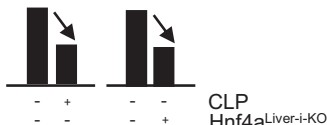

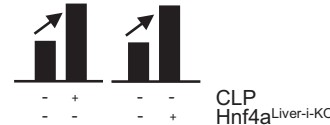

| Pathways inhibited in<br>Hnf4a<sup>flfl</sup> CLP & Hnf4a<sup>Liver-i-KO</sup> sham: Enrichr | |
|---|---|
| **Description** | **P-value** |
| Estrogen response early | $1.077e^{-5}$ |
| **Bile acid metabolism** | $4.695e^{-5}$ |
| UV response Dn | $5.669e^{-3}$ |
| **Cholesterol homeostasis** | $1.657e^{-2}$ |

| Pathways activated in<br>Hnf4a<sup>flfl</sup> CLP & Hnf4a<sup>Liver-i-KO</sup> sham: Enrichr | |
|---|---|
| **Description** | **P-value** |
| **Hypoxia** | $6.995e^{-6}$ |
| Estrogen response early | $6.995e^{-6}$ |
| **TNFα signaling via NFκB** | $3.784e^{-5}$ |
| IL-2/STAT5 signaling | $1.791e^{-4}$ |
| P53 pathway | $8.319e^{-4}$ |
| Estrogen response late | $3.329e^{-3}$ |
| E2F targets | $1.184e^{-2}$ |
| Myc targets V1 | $1.184e^{-2}$ |
| KRAS signaling Up | $1.184e^{-2}$ |
| Unfolded protein response | $1.275e^{-2}$ |
| **Apoptosis** | $1.453e^{-2}$ |
| Myc targets V2 | $3.421e^{-2}$ |
| G2-M checkpoint | $3.697e^{-2}$ |
| Adipogenesis | $3.697e^{-2}$ |
| **mTORC1 signaling** | $3.697e^{-2}$ |
| **PI3K/AKT/mTOR signaling** | $4.026e^{-2}$ |

**Figure EV3. HNF4α-dependent genes affected upon CLP function in distinct pathways.**

Enrichr pathway analysis (MSIGDB Hallmark 2020) of genes significantly downregulated or upregulated (Padj < 0.05) in Hnf4a<sup>Liver-i-KO</sup> sham relative to Hnf4a<sup>flfl</sup> (= HNF4α-dependent genes) and downregulated (Padj < 0.05, LFC < −0.8) or upregulated (Padj < 0.05, LFC > 0.8), respectively, in CLP relative to sham. *P*-values derived from Fisher's exact test (Hypergeometric test). Differential genes were identified by DESeq2 (using Wald test). The pathways we are mainly interested in, as evidenced by literature, are highlighted in bold.

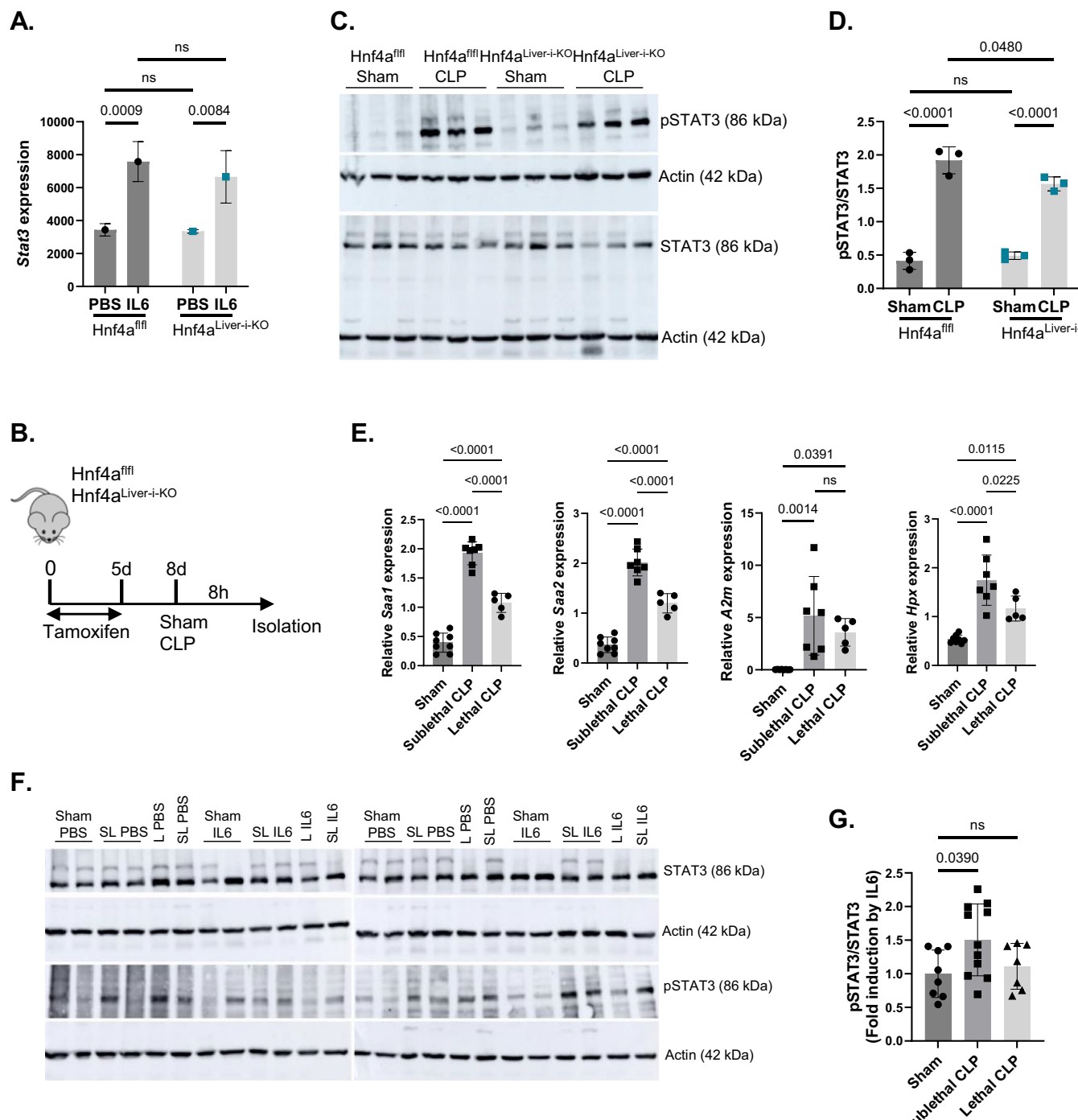

**Figure EV4. Reduced STAT3 activity and hepatic acute phase response in Hnf4a^Liver-i-KO mice and in lethal sepsis.**

(A) Hnf4a^Liver-i-KO and Hnf4a^flfl mice were treated with tamoxifen for 5 consecutive days, *i.p.* injected with hIL6 (100 μg/20 g) or PBS 3 days later, and liver was isolated 3 h later for RNA-Seq analysis. *n* = 3–4/group (biological replicates). Average normalized counts of *Stat3*. (B–D) Hnf4a^Liver-i-KO and Hnf4a^flfl mice were *i.p.* injected with tamoxifen on 5 consecutive days. Three days later, sham or CLP was performed, and livers were isolated 8 h later. (B) Experimental setup. (C, D) Western analysis of STAT3 (86 kDa) and phospho-STAT3 (Tyr705) (86 kDa) protein levels relative to actin (42 kDa). The ratio pSTAT3/STAT3 was determined as a measure for STAT3 activation. *n* = 3/group (6 female, 6 male). (E) RT-qPCR mRNA expression of *Saa1*, *Saa2*, *A2m*, and *Hpx* relative to *Hprt* and *Rpl* in livers from PBS-injected mice. *n* = 5–8/ group. (F, G) Mice were *i.p.* injected with hIL6 (100 μg/20 g) or PBS 24 h after sham or CLP, and livers were isolated 3 h later. Sublethal (=SL) vs lethal (=L) CLP were distinguished by body temperature. Western analysis of STAT3 (86 kDa) and phospho-STAT3 (Tyr705) (86 kDa) protein levels relative to actin (42 kDa). The ratio pSTAT3/STAT3 was determined as a measure for STAT3 activation (fold induction by IL6). *n* = 7–11/group. Bars: mean ± SD. Each dot represents a single biological replicate. *P*-values were analyzed with two-way ANOVA (A), (D) or one-way ANOVA (E), (G). ns: non-significant.

**A.**

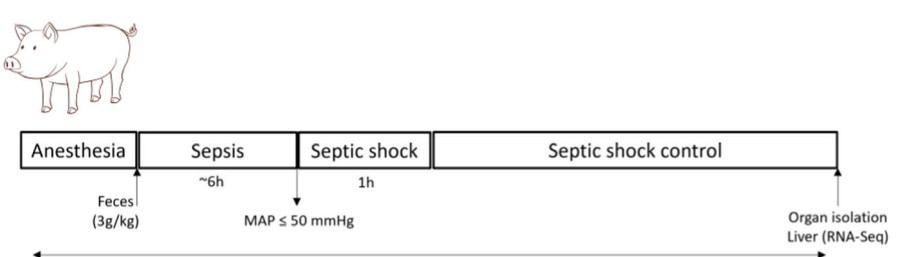

**B.**

Enrichr: downregulated genes
in pig sepsis

|   | Name | P-value |
|---|------|---------|
| 1 | **Hnf4a** | **3.083e-62** |
| 2 | Glis2 | 8.088e-11 |
| 3 | Plagl2 | 9.755e-8 |
| 4 | Creb1 | 4.782e-7 |
| 5 | Esrra | 1.992-6 |
| 6 | Prdm1 | 9.110e-6 |
| 7 | Tcof1 | 9.723e-6 |
| 8 | Yy1 | 9.567e-5 |
| 9 | Mist1 | 9.736e-5 |
| 10 | Cdx2 | 1.304e-4 |

**C.**

HOMER motif analysis: downregulated genes in pig sepsis

| Rank | Motif | Name | P-value | # Target sequences with motif | % Target sequences with motif |
|------|-------|------|---------|-------------------------------|-------------------------------|
| 1 | | HNF1b | 1e-8 | 55 | 5,24% |
| 2 | | **HNF4a** | **1e-8** | **147** | **14%** |
| 3 | | FOXA2 | 1e-7 | 168 | 16% |
| 4 | | FOXK2 | 1e-6 | 148 | 14,1% |
| 5 | | PPARa | 1e-6 | 293 | 27,9% |

**D.**

Injection of human
hepatocytes into spleen

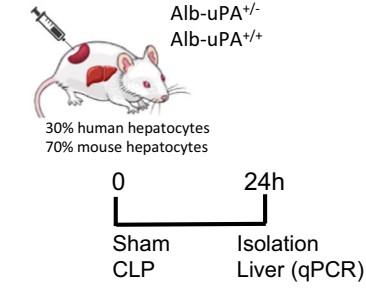

**E.**

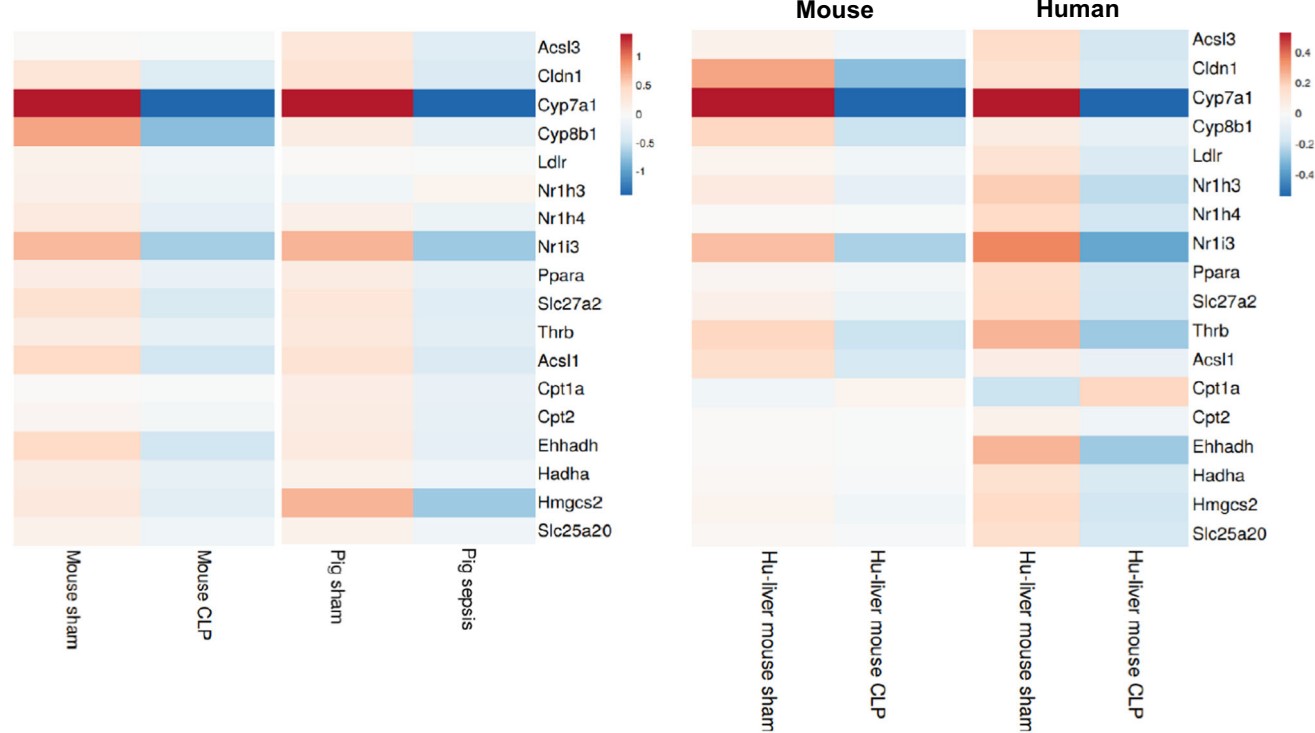

**Figure EV5.  The relevance of HNF4α loss-of-function in porcine sepsis and in septic mice with humanized liver.**

(A) Experimental setup porcine sepsis model. By intraperitoneal installation of 3 g/kg of autologous feces, the animals were permitted to develop sepsis until reaching a state of severe hypotension characterized by a mean arterial pressure (MAP) of ≤50 mmHg. Once severe hypotension, defined as a MAP ranging between 45 and 50 mmHg, was achieved, it was maintained for a duration of one hour. Around 18 h after sepsis initiation, liver samples were collected for RNA-Seq analysis ($n = 3$ sham, $n = 9$ sepsis; biological replicates). (B) Enrichr analysis of the genes downregulated (Padj < 0.05, LFC < 0) in pig sepsis. Differential genes were identified by DESeq2 (using Wald test). The upstream regulators are ordered by their enrichment *P*-value (derived from Fisher's exact test (Hypergeometric test)). (C) HOMER transcription factor motif enrichment 1000 bp upstream of TSS from the genes downregulated in pig sepsis. Differential genes were identified by DESeq2 (using Wald test). *P*-values were derived from Fisher's exact test (Hypergeometric test)). (D) At 2 weeks, Alb-uPA$^{+/+}$-SCID mice were injected with human hepatocytes into their spleens, enabling migration to the liver and repopulation of empty niches. After several weeks, mice underwent sham or CLP surgery, and livers were isolated 24 h later. (E) Heatmaps displaying the expression levels of HNF4α-dependent genes (genes differentially expressed in Hnf4a$^{Liver-i-KO}$ sham relative to Hnf4a$^{fl/fl}$ sham, with Padj < 0.05) in mouse liver 24 h post-CLP, pig sepsis liver and humanized mice liver 24 h post-CLP. We selected HNF4α target genes known to be downregulated (Padj < 0.05) in mouse liver 24 h post-CLP, focussing on the ones mentioned in the paper. Differential genes were identified by DESeq2 (using Wald test). Expression levels were quantified by RT-qPCR relative to *Hprt* and *Rpl*, and log transformed. For the humanized mice, human-specific and mouse-specific primers were used to investigate the response in human and mouse hepatocytes, respectively.

