## [Peer Review File · EMBO Molecular Medicine]

A critical role for HNF4 α in polymicrobial sepsis-associated metabolic reprogramming and death

Claude Libert, Céline Van Dender, Steven Timmermans, Ville Paakinaho, Tineke Vanderhaeghen, Jolien Vandewalle, Maarten Claes, Bruno Garcia, Bart Roman, Jan De Waele, Siska Croubels, Karolien De Bosscher, Philip Meuleman, Antoine Herpain, and Jorma Palvimo

Corresponding author: Claude Libert (claude.libert@irc.vib-ugent.be)

Review Timeline:

Submission Date:	26th Mar 24
Editorial Decision:	5th Apr 24
Revision Received:	25th Jun 24
Editorial Decision:	11th Jul 24
Revision Received:	24th Jul 24
Accepted:	13th Aug 24

Editor: Zeljko Durdevic

Transaction Report:

5th Apr 2024

Dear Prof. Libert,

Thank you for the submission of your manuscript to EMBO Molecular Medicine. We have now received feedback from the three reviewers who agreed to evaluate your manuscript. All three referees recognize potential interest of the study but also raise important concerns that should be addressed in a major revision. If you would like to discuss further the points raised by the referees, I am available to do so via email or video. Let me know if you are interested in this option.

We would welcome the submission of a revised version within three months for further consideration. Please let us know if you require longer to complete the revision.

I look forward to receiving your revised manuscript.

Yours sincerely,

Zeljko Durdevic

We require:

- 1) A .docx formatted version of the manuscript text (including legends for main figures, EV figures and tables). Please make sure that the changes are highlighted to be clearly visible.
- 2) Individual production quality figure files as .eps, .tif, .jpg (one file per figure). For guidance, download the 'Figure Guide PDF': (<https://www.embopress.org/page/journal/17574684/authorguide#figureformat>).
- 3) A .docx formatted letter INCLUDING the reviewers' reports and your detailed point-by-point responses to their comments. As part of the EMBO Press transparent editorial process, the point-by-point response is part of the Review Process File (RPF), which will be published alongside your paper.
- 4) A complete author checklist, which you can download from our author guidelines (<https://www.embopress.org/page/journal/17574684/authorguide#submissionofrevisions>). Please insert information in the checklist that is also reflected in the manuscript. The completed author checklist will also be part of the RPF.
- 5) Please note that all corresponding authors are required to supply an ORCID ID for their name upon submission of a revised manuscript.
- 6) It is mandatory to include a 'Data Availability' section after the Materials and Methods. Before submitting your revision, primary datasets produced in this study need to be deposited in an appropriate public database, and the accession numbers and

database listed under 'Data Availability'. Please remember to provide a reviewer password if the datasets are not yet public (see <https://www.embopress.org/page/journal/17574684/authorguide#dataavailability>).

13) Author contributions: You will be asked to provide CRediT (Contributor Role Taxonomy) terms in the submission system. These replace a narrative author contribution section in the manuscript.

14) A Conflict of Interest statement should be provided in the main text.

15) Every published paper now includes a 'Synopsis' to further enhance discoverability. Synopses are displayed on the journal

webpage and are freely accessible to all readers. They include a short stand first (maximum of 300 characters, including space) as well as 2-5 one-sentences bullet points that summarizes the paper. Please write the bullet points to summarize the key NEW findings. They should be designed to be complementary to the abstract - i.e. not repeat the same text. We encourage inclusion of key acronyms and quantitative information (maximum of 30 words / bullet point). Please use the passive voice. Please attach these in a separate file or send them by email, we will incorporate them accordingly.

Please also suggest a striking image or visual abstract to illustrate your article as a PNG file 550 px wide x 300-800 px high.

**** Reviewer's comments ****

Referee #1 (Comments on Novelty/Model System for Author):

Metabolic reprogramming attracts increasing attention in sepsis research. The study by van Dender et al. aims to explore the involvement of the transcription factor HNF4a in sepsis pathology. The study joins a flurry of recent investigations on HNF4a involvement in the pathogenesis of severe liver dysfunction/ disease mostly indicating correlation of disease severity with loss of function of the transcription factor. For their experiments the authors are using the whole range of contemporary approaches including liver specific knock out of the HNF4a expression in mice and analysis of the epigenetic landscape related to sepsis and HNF4a loss of function. Overall, the study significantly expands the current understanding of sepsis molecular pathology and - due to the availability of stimulants for HNF4a - the research may open a novel opportunity for pharmacological treatment of the disease.

Referee #1 (Remarks for Author):

van Dender et al. aim to explore the involvement of the transcription factor HNF4a in sepsis pathology. The authors apply a broad range of approaches including liver specific knock out of the HNF4a expression in mice and analysis of the epigenetic landscape related to sepsis and HNF4a loss of function. Overall, the study significantly expands the current understanding of sepsis molecular pathology and - due to the availability of stimulants for HNF4a - the research may open a novel opportunity for pharmacological treatment of the disease.

The authors need to address some aspects:

Major point

Both in the introduction as well in the discussion of the paper the reviewer is missing the implementation and discussion of the embedding of HNF4a in the signaling network of (diseased) hepatocytes. Putative master upstream mediators like AMPK, PGC1a, which can be pharmacologically targeted as well, have not been mentioned at all. Hence the authors discount the opportunity to present their extended view about the HNF4a related molecular pathology of sepsis. In addition, the claim of the authors on "providing new therapeutic options" remains insulated.

Minor points

I found the paper difficult to read due to the overloading with abbreviations. Possibly uncommon terms might be written out. The abbreviation LOF - as an example - has not been explained at all.

Page 4

What does "Ppara expression" mean?

Referee #2 (Comments on Novelty/Model System for Author):

Adequate but more technical info about the experiment with the humanized mouse model should be provided.

Referee #2 (Remarks for Author):

In this paper Van Dender et al. examined the mechanism of regulation of gene expression changes in the liver in response to sepsis. The authors used the model of cecal ligation and puncture (CLP). Previous studies have shown that PPARa, a key regulator of fatty acid catabolism, plays a critical role in CLP-induced lipotoxicity and tissue damage. In this paper, the authors identified HNF4a, as an upstream regulator of the sepsis induced phenotype.

The main conclusions of the paper are:

1) HNF4a protein levels are not changed in sepsis condition, but genome wide occupancy is significantly reduced. This

conclusion is well-documented by ChIP-seq experiments and extensive data analyses. A technical issue concerning the normalization procedure used for ChIP assays needs to be clarified: Did the authors use spike-in controls for quantitative assessment?

2) Chromatin structure changes were also evident to some extent on HNF4 targeted regions, which in agreement with the loss of HNF4a function.

3) Data with genetic models, demonstrate the functional role of HNF4a on sepsis-related transcriptomic profiles. From these data, the authors also conclude that HNF4a-PPAR α regulatory axis is responsible for the observed effects. This is logical, however the authors also detect similar parallel changes in a great number of other nuclear receptors in both CLP condition and HNF4a-KO livers (Fig. 5A). There is little explanation/discussion about the potential role of the other nuclear receptors.

4) The involvement of downstream processes resulting from HNF4a loss of function, e.g defects in acute phase response, are well-supported by the data.

5) The data obtained with the HNF4a agonist NCT are convincing, and support the proposal that HNF4a could be a good pharmacological target for sepsis induced liver failure.

6) The relevance of the data in other species are demonstrated by the analysis of the RNA levels of selected target genes in pig livers and humanized mouse livers. There is insufficient methodological description about the experiment in the humanized mouse liver experiment. How many cells (primary human hepatocytes?) and at what age after birth were injected? How were the injected cells isolated? How many weeks after the injections into spleens were the CLP experiments conducted?

According to Fig EV4, 30% of the hepatocytes in the liver were derived from the human cells and the RT-PCR assays were conducted with human-specific primer sets. Do the 70% of the mouse hepatocytes in these mice have lost the uPA transgene? How did the mouse hepatocyte respond to CLP? It would be important to test the RNAs from these livers with mouse-specific primers too.

7) The main weakness of the paper is that it lacks a mechanistic explanation of how sepsis affects HNF4a recruitment to its target genes. As discussed, there are many possibilities, including alternative splicing, posttranslational modifications or interactions with cofactors. Although, clarifying these possibilities is a formidable task and is probably beyond the scope of this paper, I feel that at least some of them should be tested here. For example, RT-PCR analysis for the known splicing variants of HNF4a and an in vitro DNA-binding assay (band-shift), could be easily performed and exclude or prove changes in isoform distribution or DNA binding affinity.

Referee #3 (Comments on Novelty/Model System for Author):

In addition to several mouse models, also pigs and humanized mice are used. This represents a very high standard.

Referee #3 (Remarks for Author):

In this article Libert et al. report a role for HNF4 α in polymicrobial sepsis-associated metabolic reprogramming and death. They highlight an important connection between HNF4 α downregulation during sepsis, with subsequent PPAR downregulation, which prevents an adequate response towards IL6. Employing the HNF4 α agonist NCT, the authors show a protective role for HNF4 α in sepsis, which might open up new therapeutic approaches. The manuscript comprises state-of-the art methodology.

I would like to raise a few concerns:

- Data in figure 5I do not find an altered lipid droplet volume after CLP, which was rather pronounced in figure 7J. Similarly, blood FFA were not significantly increased in 5H, but distinctly in figure 7I. Do the authors have an explanation for this discrepancy? Can this be explained by the presence of DMSO (which can have lots of biological effects even at low concentrations)? Since LPS treatment as a simplified sepsis model does increase hepatic lipid content (doi: 10.1016/j.imbio.2017.01.003): were there differences in the bacterial load or other differences in the two experiments?
- The authors study the response of livers towards IL6 treatment. Since IL6 is important for inflammation-associated hepatocyte lipid storage via STAT3: What is the action of IL6 administration on lipid metabolism (lipid content in livers, FFA in blood)?
- With n=3 for some of the data, the number of mice investigated seems rather low. Did biostatistics calculations before the experiments suggest that this is a number expected to be sufficient?
- In figure 2A extra sham controls were used for the two different time points. In figure 2B there is only one sham group. How long was these animals' sham treatment?
- The authors mention that male and female mice were used for experiments. With infectious and inflammatory diseases exhibiting differences between males and females, it would be important to mention for each figure / result, the number of animals for each sex. Did the authors observe any differences between sexes?

- The authors should introduce each abbreviation before their first usage.
- Since Dr. Meuleman is a co-author on the manuscript it is not clear why he is mentioned as the person "generously" providing animals in the Methods section.

Point-by-point response to the comments of the reviewers.**Referee #1 (Comments on Novelty/Model System for Author):**

Metabolic reprogramming attracts increasing attention in sepsis research. The study by van Dender et al. aims to explore the involvement of the transcription factor HNF4a in sepsis pathology. The study joins a flurry of recent investigations on HNF4a involvement in the pathogenesis of severe liver dysfunction/ disease mostly indicating correlation of disease severity with loss of function of the transcription factor. For their experiments the authors are using the whole range of contemporary approaches including liver specific knock out of the HNF4a expression in mice and analysis of the epigenetic landscape related to sepsis and HNF4a loss of function. Overall, the study significantly expands the current understanding of sepsis molecular pathology and - due to the availability of stimulants for HNF4a - the research may open a novel opportunity for pharmacological treatment of the disease.

Answer: we thank the reviewer for the supportive comments.

Referee #1 (Remarks for Author):

van Dender et al. aim to explore the involvement of the transcription factor HNF4a in sepsis pathology. The authors apply a broad range of approaches including liver specific knock out of the HNF4a expression in mice and analysis of the epigenetic landscape related to sepsis and HNF4a loss of function. Overall, the study significantly expands the current understanding of sepsis molecular pathology and - due to the availability of stimulants for HNF4a - the research may open a novel opportunity for pharmacological treatment of the disease.

The authors need to address some aspects:

Major point

Both in the **introduction as well in the discussion** of the paper the reviewer is missing the implementation and discussion of the embedding of HNF4a in the signaling network of (diseased) hepatocytes. Putative **master upstream mediators like AMPK, PGC1a, which can be pharmacologically targeted** as well, have not been mentioned at all. Hence the authors discount the opportunity to present their extended view about the HNF4a related molecular pathology of sepsis. In addition, the claim of the authors on "providing new therapeutic options" remains insulated.

Answer: The reviewer has a valid point. Many upstream regulators of HNF4 α have been described in literature and could be worthwhile to look further into to better understand the mechanism of the observed HNF4 α loss-of-function in sepsis, as well as to provide additional therapeutic options besides NCT. The data from the HNF4 α ChIP-Seq and HNF4 α DNA binding assay (**Figure EV3**) show that HNF4 α DNA binding is reduced, in general, during sepsis. Given that there is no observed difference in the total protein levels of HNF4 α , the percentage HNF4 α -positive nuclei, and the relative abundance of HNF4 α isoforms between sham and CLP (**Figure EV3**), the most logical explanation is that HNF4 α is either modified or has reduced interaction with coactivators during sepsis. Firstly, several kinases have been

described to phosphorylate and thereby inhibit HNF4 α , including protein kinase A (Viollet et al, 1997), protein kinase C (Sun et al, 2007), ERK1/2 kinase, AMPK and src kinase. Src kinase has been shown to phosphorylate HNF4 α , leading to its cytoplasmic translocation and degradation (in a model of partial hepatectomy) (Huck et al, 2019). However, we believe that this is not relevant for sepsis as our data suggest that HNF4 α remains in the nucleus. AMPK regulates HNF4 α transcriptional activity by inhibiting dimer formation and decreasing protein stability (Hong et al, 2003). Despite we do not observe a difference in the total protein levels of HNF4 α , AMPK activation could still impact HNF4 α binding during sepsis. However, AMPK KO mice are more sensitive to sepsis, while AMPK activation offers protection (Kikuchi et al, 2020; Jin et al, 2020). Interestingly, HNF4 α phosphorylation by ERK1/2 impacts the strength of DNA binding rather than its location, which we also observe in our data (Vetö et al, 2017). However, the inhibition of ERK1/2 with SCH772984 also enhances survival in both the LPS and CLP models, which was attributed to a reduced immune response and platelets activation, as well as upregulation of the extracellular matrix organization and retinoic acid signalling pathways (Kopczynski et al, 2021). These data suggest that by targeting upstream regulators such as AMPK and ERK1/2, a far broader array of downstream targets will be impacted in comparison to specifically targeting HNF4 α . Secondly, coactivators of HNF4 α include NCOA1, NCOA2, NCOA3, EP300, CREBBP, GRIP1 and PPARGC1a. According to the literature, interaction between HNF4 α and its coactivators is essential for locking its active conformation and thereby may have impact on its DNA binding capacity (Duda et al, 2004). While no interactomics studies for HNF4 α in sepsis have been done so far, we know that all coactivators, except Ppargc1a, show comparable expression levels (at the mRNA level) between sham and CLP at 6h (**Figure EV1**). PGC1 α not only interacts with HNF4 α but also with PPAR α to regulate mitochondrial β -oxidation (Vega et al, 2000). All of these observations suggest that PGC1 α could be an interesting therapeutic target for sepsis. Indeed, the PGC1 α activator ZLN005 has already been shown to improve survival in polymicrobial sepsis, but its effect on HNF4 α remains unknown (Suzuki et al, 2023). Thirdly, TGF β was described as a key upstream regulator in alcoholic hepatitis by inducing Hnf4a expression from the P2 promoter (instead of P1) in human hepatocytes (Argemi et al, 2019). However, we didn't observe a difference in the relative abundance of HNF4 α isoforms between sham and CLP.

In conclusion, various mechanisms regulate HNF4 α expression and activity in healthy & diseased hepatocytes. In the revised paper, we have outlined some of these mechanisms in the introduction and have expanded on those that might explain our observations of HNF4 α loss-of-function in sepsis, potentially providing additional therapeutic targets, in the new discussion.

Figure EV1: Expression of HNF4 α coactivators in liver during sepsis.

RNA-Seq analysis on liver from mice 6h post-CLP or sham surgery. Expression of several HNF4a coactivators is plotted as log normalized counts. Data is analysed by simple linear regression.

Minor points

I found the paper difficult to read due to the overloading with **abbreviations**. Possibly uncommon terms might be written out. The abbreviation LOF - as an example - has not been explained at all.

Answer: *We understand the comment of the reviewer. Therefore, we removed uncommon abbreviations or those not frequently referred to, such as AAs (amino acids), WAT (white adipose tissue), and AP (acute phase). However, to reduce the word count, we kept abbreviations such as FFA (free fatty acids), APR (acute phase response), LOF (loss of function) and principle component analysis (PCA), but redefined them in each section of the paper to improve readability.*

Page 4

What does "Ppara expression" mean?

Answer: *"Recent studies support that HNF4a acts as key activator of Ppara expression in the liver by binding to the DR1 motif within its gene promoter." In this context, "expression" specifically refers to transcription. We have revised this point in the manuscript to improve clarity, and have written Ppara mRNA expression.*

Referee #2 (Comments on Novelty/Model System for Author):

Adequate but more technical info about the experiment with the humanized mouse model should be provided.

Answer: *our apologies if this was not clear in the paper. The humanized mouse model (provided by Prof. Meuleman) was described in detail in (Meuleman et al, 2005). The mice are generated by transplanting homozygous Alb-uPA^{+/+}-SCID mice (which suffer from spontaneous and chronic death of hepatocytes) with 0.7×10^6 cryopreserved primary human hepatocytes (donor C342, Lonza), via intrasplenic injection. The human albumin concentration in plasma was determined 6 weeks after transplantation by ELISA (Bethyl Laboratories, Montgomery, Texas, United States) and was used as a marker of liver chimerism. Such mice underwent sham or CLP surgery, after which their livers were isolated 24h later for qPCR analysis. More information about the humanized mouse model can be found in (Meuleman et al, 2005).*

This info has now been included in the methods section of the revised paper.

In this paper Van Dender et al. examined the mechanism of regulation of gene expression changes in the liver in response to sepsis. The authors used the model of cecal ligation and puncture (CLP). Previous studies have shown that PPAR α , a key regulator of fatty acid catabolism, plays a critical role in CLP-induced lipotoxicity and tissue damage. In this paper, the authors identified HNF4a, as an upstream regulator of the sepsis induced phenotype.

The main conclusions of the paper are:

1) HNF4a protein levels are not changed in sepsis condition, but genome wide occupancy is significantly reduced. This conclusion is well-documented by ChIP-seq experiments and extensive data analyses. A technical issue concerning the normalization procedure used for ChIP assays needs to be clarified: **Did the authors use spike-in controls for quantitative assessment?**

Answer: *no, we didn't include spike-in controls for normalization. We did use an IgG antibody (derived from the same species as the HNF4a antibody, i.e., mouse) as a control and normalized our data against an input sample taken before sonication. For quantitative assessment, we utilized several independent replicates. If a consistent sham-vs-CLP difference is evident across all individual replicates (sham rep1-4 vs CLP rep1-4), this highlights its biological relevance. We clarified in the materials and methods section of the revised paper that we did not use spike-in controls.*

2) Chromatin structure changes were also evident to some extent on HNF4 targeted regions, which in agreement with the loss of HNF4a function.

3) Data with genetic models, demonstrate the functional role of HNF4a on sepsis-related transcriptomic profiles. From these data, the authors also conclude that HNF4a-PPARa regulatory axis is responsible for the observed effects. This is logical, however the authors also detect similar parallel changes in a great number of other nuclear receptors in both CLP condition and HNF4a-KO livers (Fig. 5A). **There is little explanation/discussion about the potential role of the other nuclear receptors.**

Answer: *indeed, besides the interaction between HNF4a and PPARa in sepsis, we found that HNF4a depletion impaired the expression of many other nuclear receptors, among which retinoid X receptor-α (RXRα), farnesoid X receptor (FXR), constitutive androstane receptor (CAR), androgen receptor (AR) and liver X receptor-α (LXRα). These data indicate that hepatic HNF4a loss-of-function has a broad downstream impact in sepsis. As mentioned in the discussion of the paper, reduced expression of FXR, LXRα and RXRα in sepsis has already been described (Wang et al, 2011a; Hao et al, 2017; Chen et al, 2007). FXR plays a critical role in regulating many genes involved in bile acid metabolism. Critically ill sepsis patients frequently display cholestatic hepatic dysfunction, which is associated with increased mortality. Cholestasis may further result in impaired bile secretion and subsequent accumulation of bile acids in the circulation (Horvatits et al, 2017). Experimental cholestasis sensitizes mice to LPS-induced sepsis, while cholestyramine, a bile acid sequestrant, provides partial protection. Furthermore, FXR overexpression restored the expression of FXR target genes, reduced serum bile acid levels and thereby partially protected against LPS-induced septic shock, while FXR KO mice are more sensitive (Hao et al, 2017). LXRα regulates genes involved in (lipid) metabolism, inflammation and apoptosis. Septic mice or rats treated with the LXR agonist GW3965 were partially protected, showing decreased multi-organ injury and reduced inflammatory cytokine levels, while LXRα-deficient mice exhibited enhanced liver injury (Zhang et al, 2021; Wang et al, 2011b). Although the xenobiotic receptor CAR has never been investigated in sepsis, many hepatic and intestinal cytochrome P450 enzymes and drug transporters have been found to be downregulated in sepsis (Lv & Huang, 2020). All these data suggest that targeting HNF4a will affect multiple nuclear receptors besides the HNF4a-PPARa regulatory axis, thereby potentially improving various sepsis-associated features such as hepatic steatosis, cholestasis, apoptosis, and alterations in drug metabolism, all frequently observed in sepsis patients.*

This elaboration on the role of other nuclear receptors downstream of the observed HNF4a loss-of-function in sepsis is now included in the discussion of the revised paper.

4) The involvement of downstream processes resulting from HNF4a loss of function, e.g defects in acute phase response, are well-supported by the data.

5) The data obtained with the HNF4a agonist NCT are convincing, and support the proposal that HNF4a could be a good pharmacological target for sepsis induced liver failure.

6) The relevance of the data in other species are demonstrated by the analysis of the RNA levels of selected target genes in pig livers and humanized mouse livers. There is insufficient methodological description about the experiment in the humanized mouse liver experiment. How many cells (primary human hepatocytes?) and at what age after birth were injected? How were the injected cells isolated? How many weeks after the injections into spleens were the CLP experiments conducted? According to Fig EV4, 30% of the hepatocytes in the liver were derived from the human cells and the RT-PCR assays were conducted with human-specific primer sets. **Do the 70% of the mouse hepatocytes in these mice have lost the uPA transgene?**

How did the mouse hepatocyte respond to CLP? It would be important to test the RNAs from these livers with mouse-specific primers too.

***Answer:** the repopulation of human hepatocytes in the liver is determined by the ratio of human vs mouse albumin in the blood of the mice. In principle, all mouse hepatocytes are homozygous for the uPA transgene, meaning that the regenerating mouse hepatocytes only derive from proliferation and progenitor cell differentiation. Because of the constant turnover of mouse hepatocytes, a certain percentage of mouse albumin will always be measured in the blood, but it doesn't necessarily indicate that these hepatocytes are completely healthy as they still express the uPA transgene. According to the scientist, who generates the mice (and has been working with them for > 20 years, Dr. Meuleman), over time, due to the high turnover rate in the liver, it is indeed possible that some mouse hepatocytes may lose the transgene, resulting in a healthy liver nodule. However, in our case, this should not occur since we used young animals. We made this more clear in the result part of the revised paper.*

*To investigate the response of mouse hepatocytes from uPA transgenic mice to CLP, we conducted qPCR using mouse-specific primers for the same PPAR α and HNF4 α target genes examined previously. We observed a similar trend as seen in human hepatocytes, normal CLP mice, and septic pigs (**Figure EV2**). These data were added to **Figure EV5D** of the revised paper.*

Figure EV2: **The relevance of HNF4 α loss-of-function in porcine sepsis and in septic mice with humanized liver.**

Heatmaps displaying the expression levels of HNF4 α -dependent genes (genes differentially expressed in *Hnf4a*^{Liver-i-KO} sham relative to *Hnf4a*^{fl/fl} sham, with *Padj* < 0.05) in mouse liver 24h post-CLP, pig sepsis liver and humanized mice liver 24h post-CLP. We selected HNF4 α target genes known to be downregulated (*Padj* < 0.05) in mouse liver 24h post-CLP, focussing on the ones mentioned in the paper. Expression levels were quantified by RT-qPCR relative to *Hprt* and *Rpl*, and log transformed. For the humanized mice, human-specific and mouse-specific primers were used to investigate the response in human and mouse hepatocytes, respectively.

7) The main weakness of the paper is that it lacks a mechanistic explanation of how sepsis affects HNF4 α recruitment to its target genes. As discussed, there are many possibilities, including alternative splicing, posttranslational modifications or interactions with cofactors. Although, clarifying these possibilities is a formidable task and is probably beyond the scope of this paper, I feel that at least some of them should be tested here. For example, **RT-PCR analysis for the known splicing variants of HNF4 α and an in vitro DNA-binding assay** (band-shift), could be easily performed and exclude or prove changes in isoform distribution or DNA binding affinity.

Answer: we understand the concern of the reviewer. To investigate the changes of DNA binding of HNF4 α in sepsis, we performed ATAC-Seq and ChIP-Seq for histone modifications H3K4me3 and H3K27ac. The results indicated that chromatin remodelling and H3K27 acetylation changes were downstream effects of the observed HNF4 α loss-of-function in sepsis. Alternative mechanisms such as alternative splicing, posttranslational modifications or changes in the interactome were considered. Unlike the human *Hnf4a* gene with its 17 transcripts, the mouse *Hnf4a* gene has 5 transcripts, 4 of which are protein-coding. RNA-Seq data revealed only 3 detectable transcripts in the liver. Due to the high sequence similarity, isoform-specific primers could not be designed. We thus compared the relative abundance of these 3 isoforms between sham and CLP conditions using in-house paired-end RNA-Seq data 8 hours post-CLP. However, no significant difference could be observed (**Figure EV3A**). To further understand the impact of sepsis on the DNA binding capacity of HNF4 α , we performed an in vitro DNA binding assay using an ELISA-based method. In this method, we added nuclear lysate to a dsDNA oligo-coated plate. The signal intensity (OD450-OD570) was proportional to the amount of HNF4 α / γ bound

to the plate. Given that HNF4 α is significantly more expressed in the liver compared to HNF4 γ , the contribution of HNF4 γ to the measured signal is considered very minimal. To validate the assay, we first added nuclear lysate (10 μ g) from liver-specific HNF4 α KO liver samples. A significant reduction in signal intensity could be observed (Figure EV3B). Similarly, when lysate (50 μ g) from sham and CLP samples collected 8 hours post-CLP was added, a significant decrease in HNF4 α DNA binding was observed (Figure EV3C). These data suggest that HNF4 α is likely modified in sepsis, potentially through posttranslational modifications or altered interactions with cofactors that affect its DNA binding capacity. However, this is beyond the scope of the present study and will be investigated further in a follow-up paper. Studying PTMs and protein-protein interactions will be done by RIME, followed by functional estimation of the relevance of the findings (including HNF4 α mutagenesis of PTM sites, work using enzyme inhibitors, e.g. of acetyl transferases, mutagenesis of co-factors). Right now, we are at the beginning of this research, and have optimized the RIME protocol. But we are still far from ready to report on any finding. This will take at least another year, maybe two, so we consider this as follow-up of this first paper, the purpose of which is to report on the very strong links between sepsis and hepatic HNF4 α LOF during disease progression.

We have included these results in the revised paper (Figure EV1). Based on these new data and the results already discussed and relevant literature, we will propose potential mechanisms responsible for the loss of HNF4 α function in sepsis. We will focus on molecules that can be pharmaceutically targeted. Examples include phosphorylation mediated by AMPK or ERK1/2 kinases or decreased interaction with PGC1 α .

Figure EV3: **Absence of alternative splicing with reduced HNF4 α DNA binding in septic liver.** (A) Relative abundance of Hnf4a transcripts (Hnf4a-201, Hnf4a-202, and Hnf4a-203) in CLP relative to sham, expressed as log₂(TPM) percentages, derived from in-house paired-end RNA-Seq data 8h post-CLP. (B-C) OD450-OD570 represents the strength of HNF4 α binding to dsDNA oligos immobilized on a plate using nuclear lysates from tamoxifen-injected Hnf4 α ^{fl/fl} and Hnf4 α ^{Liver-i-KO} mice (B) or sham and CLP mice 8h post-CLP (C). Error bars represent SD. Central lines represent mean. Each dot represents a single biological replicate. P-values were analysed with two-way ANOVA (A) or unpaired t-test (B-C). ns: nonsignificant, *** $p \leq 0.001$, **** $p \leq 0.0001$.

Referee #3 (Comments on Novelty/Model System for Author):

In addition to several mouse models, also pigs and humanized mice are used. This represents a very high standard.

Referee #3 (Remarks for Author):

In this article Libert et al. report a role for HNF4 α in polymicrobial sepsis-associated metabolic reprogramming and death. They highlight an important connection between HNF4 α downregulation during sepsis, with subsequent PPAR α downregulation, which prevents an adequate response towards IL6. Employing the HNF4 α agonist NCT, the authors show a protective role for HNF4 α in sepsis, which might open up new therapeutic approaches. The manuscript comprises state-of-the-art methodology.

I would like to raise a few concerns:

- Data in figure 5I do not find an altered lipid droplet volume after CLP, which was rather pronounced in figure 7J. Similarly, blood FFA were not significantly increased in 5H, but distinctly in figure 7I. Do the authors have an explanation for this discrepancy? Can this be explained by the presence of DMSO (which can have lots of biological effects even at low concentrations)? Since LPS treatment as a simplified sepsis model does increase hepatic lipid content (doi: 10.1016/j.imbio.2017.01.003): **were there differences in the bacterial load or other differences in the two experiments?**

Answer. We thank the reviewer for the very supportive and good comments. We measured the bacterial load only in the NCT experiment, not in the HNF4aKO experiment. When plotting plasma FFA values and hepatic lipid content for the Hnf4 $\alpha^{fl/fl}$ mice only, we observed a significant increase after CLP, which is consistent with the literature (**Figure EV4**, for the attention of the reviewer) (Van Wyngene et al, 2020). However, this significance was lost in ANOVA due to correction for multiple testing. The FFA levels post-CLP were comparable to those in DMSO-treated CLP. Interestingly, the sham condition in Hnf4 $\alpha^{fl/fl}$ mice showed higher FFA levels, likely due to tamoxifen being dissolved in an oil solution. The same can be said about the hepatic lipid content. The values for Hnf4 $\alpha^{fl/fl}$ CLP and DMSO CLP are rather similar, but the Hnf4 $\alpha^{fl/fl}$ sham condition already shows some lipid accumulation in the liver (due to the oil solution that was injected).

Small remark: For the HNF4aKO experiment, blood FFA levels were originally normalized against sham. To maintain consistency between experiments, we have now plotted the absolute levels in the revised paper (Figure 5H).

Figure EV4: Significantly increased plasma FFA levels and hepatic lipid content 8h after CLP, relative to sham, in Hnf4 $\alpha^{fl/fl}$ mice. Hnf4 $\alpha^{fl/fl}$ mice were injected with tamoxifen for 5 consecutive days. Three days later, CLP and sham surgeries were performed, and blood and liver were collected 8h afterward. Plasma FFA levels were quantified with a kit. Volume of hepatic lipid droplets (represented by voxel counts), relative to liver tissue volume, was calculated for

each z-stack and averaged over all z-stacks per section. Lipid droplets were stained with LipidTox. Bars: mean \pm SD. Each dot represents a single biological replicate. P-values were analysed with unpaired t-test. * $p \leq 0.05$, ** $p \leq 0.01$.

• The authors study the response of livers towards IL6 treatment. Since IL6 is important for inflammation-associated hepatocyte lipid storage via STAT3: **What is the action of IL6 administration on lipid metabolism (lipid content in livers, FFA in blood)?**

Answer: *this is a very good (but difficult) question of the reviewer. We did our best to address it. IL6 has been described in the literature to affect both the degradation and the synthesis of fatty acids in the liver, depending on the context (Gavito et al, 2016). Therefore, IL6 administration has shown benefits in some fatty liver models, such as alcohol liver disease, but not in others (Hong et al, 2004). IL6KO mice exhibit increased hepatic steatosis, supporting the protective role of IL6 (Kroy et al, 2010). However, elevated plasma IL6 levels in chronic conditions such as fatty liver disease (and sepsis) argue against its protective role (Glund & Krook, 2008). Additionally, the administered IL6 dose also seems to play a role. Chronic low-dose IL6 (ng/g range) in IL6KO mice fed with a high-fat diet increased hepatic steatosis by enhancing lipogenesis, linked to increased STAT3 and AMPK activation (Vida et al, 2015). Conversely, chronic high-dose IL6 ($\mu\text{g/g}$ range) reduced steatosis in 3 models of fatty liver disease (leptin deficient ob/ob mice, ethanol-fed mice and high-fat diet mice) by improving mitochondrial fatty acid β -oxidation, mediated by upregulating PPAR α , and increasing hepatic export of triglycerides and cholesterol. Notably, only long-term (10 days) administrations, not short-term (1 day), could offer protection (Hong et al, 2004). In our paper, we administered a single shot of IL6 at 5 $\mu\text{g/g}$ to the mice, which significantly upregulated acute phase proteins in the blood after 12h and provided protection against sepsis. Given the acute nature of the CLP model and our focus on the early phase of sepsis, chronic IL6 administration was not feasible. Therefore, and in light of our findings regarding protection, we studied lipid metabolism following a single IL6 injection at 5 $\mu\text{g/g}$. Mice underwent sham or CLP surgery 12h after this injection, and 24h later, blood and liver samples were collected to measure hepatic STAT3 activity by Western blot analysis, FFA levels in the blood, and lipid content in the liver by LipidTox staining. However, pre-administering IL6 to septic mice significantly increased STAT3 activity in the liver but didn't affect lipid accumulation in the blood and liver (**Figure EV5**). The difference between our results and those in the literature could be related to our single-dose administration approach. Additionally, while sepsis shares some aspects with the high-fat diet model, it remains fundamentally different. In general, our results indicate that the observed protection against sepsis cannot be attributed to an improvement in FFA levels in the blood or lipid content in the liver, but is likely more related to the increase in acute phase proteins (and hence the impact on regeneration). As discussed in the paper, several acute phase proteins have already shown protection in sepsis and sepsis-like models (Libert et al, 1994; Hochepped et al, 2000; Van Molle et al, 1997; Dalli et al, 2014; Libert et al, 1996; Janz et al, 2013; Larsen et al, 2010) and are known to play physiological roles in hepatocellular regeneration, in addition to their anti-inflammatory and/or antimicrobial functions (Moshage, 1997; Streetz et al, 2000; Cressman et al, 1996; Greenbaum et al, 1998). We believe that HNF4 α loss-of-function in sepsis has 2 major consequences: 1) reduced expression and activity of multiple nuclear receptors, leading to loss of metabolic functions such as PPAR α and mitochondrial fatty acid oxidation; and 2) impaired activation of the acute phase response downstream of IL6, both contributing to lethality. By targeting HNF4 α with NCT, we effectively addressed both aspects and improved survival. With IL6, only 19% of the mice were protected, whereas with NCT, we achieved approximately 30% protection. These additional data strengthen us in our belief that both aspects are critical for survival.*

In the discussion of the paper, we have already linked the IL6-mediated protection to an improved acute phase response. Since we did not observe an effect of IL6 administration on lipid metabolism, we decided to not include these data in the revised paper. To meet the request of the reviewer, we have decided, however, to mention the experiments in the discussion claiming that we found protection by

IL6, clearly related to increase of acute phase proteins, but with no obvious impact on fat metabolism during sepsis.

Figure EV5: IL6 administration increases STAT3 activity in the liver without affecting lipid accumulation in the blood and liver. Mice were i.p. injected with hIL6 (5 mg/kg) or PBS 12h before sham or CLP surgery. Liver and blood samples were collected 24h post-surgery. **(A)** Western analysis of STAT3 (86 kDa) and phosphor-STAT3 (Tyr705) (86 kDa) protein levels relative to actin (42 kDa). The ratio pSTAT3/STAT3 was determined as a measure for STAT3 activation (fold induction by IL6). **(B)** Plasma FFA levels 24h after CLP. **(C)** Volume of hepatic lipid droplets (represented by voxel counts), relative to liver tissue volume, was calculated for each z-stack and averaged over all z-stacks per section. Lipid droplets were stained with LipidTox. Bars: mean \pm SD. Each dot represents a single biological replicate. P-values were analysed with two-way ANOVA. ns: nonsignificant, * $p \leq 0.05$, ** $p \leq 0.01$.

• With n=3 for some of the data, the number of mice investigated seems rather low. Did biostatistics calculations before the experiments suggest that this is a number expected to be sufficient?

Answer: we acknowledge the reviewer's concern. For multi-omics analyses (RNA-Seq, ChIP-Seq, ATAC-Seq) an n-value of 3-4 is adequate to derive biologically relevant insights. However, in response to this relevant comment, for some experiments such as HNF4 α immunohistochemistry, qPCR of GW7647-induced PPAR α target genes in liver-induced HNF4 α KO mice, and STAT3/pSTAT3 Western blotting in sublethal and lethal CLP IL6 samples, we have increased the n-values to enhance statistical robustness. The same conclusions could be drawn as before: there is no difference in the percentage of HNF4 α -positive nuclei between sham and CLP conditions, a significant decrease in the KO, reduced PPAR α activity in absence of hepatic HNF4 α and reduced STAT3 activation in lethal compared to sublethal CLP in response to IL6 (Figure EV6).

Figure EV6: No difference in HNF4 α -positive nuclei between sham and CLP, with significant decrease in HNF4 α KO, reduced PPAR α activity in absence of hepatic HNF4 α and reduced STAT3 activation in lethal vs. sublethal CLP in response to IL6. (A) Immunofluorescent images of liver 6h, 24h and 48h after CLP or sham, or from tamoxifen-injected *Hnf4a*^{fl/fl} and *Hnf4a*^{Liver-i-KO} mice, stained with HNF4 α antibody (red) and DAPI (blue). Scale bar = 20 μ m. Percentage HNF4 α positive nuclei was quantified. **(B)** Mice were i.p. injected with hIL6 (5 mg/kg) or PBS 24h after sham or CLP, and livers were isolated 3h later. Sublethal vs. lethal CLP were distinguished by body temperature. Western analysis of STAT3 (86 kDa) and phospho-STAT3 (Tyr705) (86 kDa) protein levels relative to actin (42 kDa). The ratio pSTAT3/STAT3 was determined as a measure for STAT3 activation (fold induction by IL6). **(C)** PPAR α agonist GW7647 or vehicle were injected i.p. in *Hnf4a*^{Liver-i-KO} and *Hnf4a*^{fl/fl} mice three days after tamoxifen treatment. Livers were isolated for qPCR 4h later. RT-qPCR mRNA expression of *Ppara* and several PPAR α target genes relative to *Gapdh* and *Hprt*. Bars: mean \pm SD. Each dot represents a single biological replicate. P-values were analysed with two-way ANOVA. ns: nonsignificant, * $p \leq 0.05$, ** $p \leq 0.01$, *** $p \leq 0.001$, **** $p \leq 0.0001$.

- In figure 2A extra sham controls were used for the two different time points. In figure 2B there is only one sham group. **How long was these animals' sham treatment?**

Answer: 50% of the sham-operated mice were isolated 8h after CLP, and the remaining 50% were isolated 24h after CLP. Since we did not observe any difference in HNF4 α protein levels between the timepoints in sham mice, we collectively referred to them as sham-operated mice.

- The authors mention that male and female mice were used for experiments. With infectious and inflammatory diseases exhibiting differences between males and females, it would be important to mention for each figure / result, the number of animals for each sex. **Did the authors observe any differences between sexes?**

Answer: for experiments involving wildtype C57BL6 mice, only male mice were used because of 'pragmatic reasons', meaning that over the past decades that we have been applying the CLP model, males have appeared more consistent in their response compared to females, presumably because the latter may have hormonal fluctuations depending on the menstruation cycle. But this does not imply that female mice should be avoided or are worthless. Only results pertaining to HNF4 α KO mice included both male and female mice. In these experiments, we aimed to ensure an equal number of animals of the same sex per condition to mitigate variations in the response to sepsis. By doing so, we ensure that any differences observed between the conditions are not influenced by sex-related factors. For the characterization experiment with the KO, more male mice were used. As we show in the figures here, for the attention of the reviewer, it is clear that indeed the female mice were slightly less responsive (at the level of body T) to the CLP procedure compared to the male mice, but avoiding them would not be scientifically correct, and would not change the conclusion of the experiment. Furthermore, in the HNF4 α KO RNA-Seq, the samples clustered clearly together per condition in the principal component analysis (PCA) plot, regardless of sex, suggesting that their response (at the level of gene expression) in the knockout and in CLP is quite similar.

The number of animals for each sex was included in the legend of each figure in the revised paper.

B

Figure EV7: Sex has a small effect on body temperature response to CLP, but not on gene expression. *Hnf4a*^{fl/fl} and *Hnf4a*^{Liver-i-KO} mice were injected with tamoxifen for 5 consecutive days. Three days later, CLP and sham surgeries were performed, and blood and liver were collected 8h afterward. **(A)** Body temperature 8h after sham or CLP was measured for all mice, irrespective of gender, as well as for male mice only and female mice only. **(B)** PCA analysis of HNF4aKO RNA-Seq on livers collected 8h after sham or CLP.

- The authors should introduce each abbreviation before their first usage.

Answer: Thank you. Every abbreviation was checked and reintroduced in each section of the revised paper to enhance readability.

- Since Dr. Meuleman is a co-author on the manuscript it is not clear why he is mentioned as the person "generously" providing animals in the Methods section.

Answer: Thank you. We have removed this sentence in the materials and methods part of the revised paper and added some information about the generation of these mice.

References

- Argemi J, Latasa MU, Atkinson SR, Blokhin IO, Massey V, Gue JP, Cabezas J, Lozano JJ, Van Booven D, Bell A, *et al* (2019) Defective HNF4 α -dependent gene expression as a driver of hepatocellular failure in alcoholic hepatitis. *Nat Commun* 10
- Chen YH, Hong IC, Kuo KK, Hsu HK & Hsu C (2007) Role of retinoid-X receptor- α in the suppression of rat bile acid coenzyme A-amino acid N-acyltransferase in liver during sepsis. *Shock* 28: 65–70
- Cressman DE, Greenbaum LE, DeAngelis RA, Ciliberto G, Furth EE, Poli V & Taub R (1996) Liver failure and defective hepatocyte regeneration in interleukin-6- deficient mice. *Science (1979)* 274: 1379–1383
- Dalli J, Norling L V., Montero-Melendez T, Canova DF, Lashin H, Pavlov AM, Sukhorukov GB, Hinds CJ & Perretti M (2014) Microparticle alpha-2-macroglobulin enhances pro-resolving responses and promotes survival in sepsis. *EMBO Mol Med* 6: 27–42
- Duda K, Chi YI & Shoelson SE (2004) Structural basis for HNF-4 α activation by ligand and coactivator binding. *Journal of Biological Chemistry* 279: 23311–23316
- Gavito AL, Bautista D, Suarez J, Badran S, Arco R, Pavón FJ, Serrano A, Rivera P, Decara J, Cuesta AL, *et al* (2016) Chronic IL-6 administration desensitizes IL-6 response in liver, causes hyperleptinemia and aggravates steatosis in diet-induced-obese mice. *PLoS One* 11
- Glund S & Krook A (2008) Role of interleukin-6 signalling in glucose and lipid metabolism. In *Acta Physiologica* pp 37–48.
- Greenbaum LE, Li W, Cressman DE, Peng Y, Ciliberto G, Poli V & Taub R (1998) CCAAT Enhancer-binding Protein Is Required for Normal Hepatocyte Proliferation in Mice after Partial Hepatectomy. *J Clin Invest* 102: 996–1007
- Hao H, Cao L, Jiang C, Che Y, Zhang S, Takahashi S, Wang G & Gonzalez FJ (2017) Farnesoid X Receptor Regulation of the NLRP3 Inflammasome Underlies Cholestasis-Associated Sepsis. *Cell Metab* 25: 856-867.e5
- Hochepped T, Van Molle W, Berger FG, Baumann H & Libert C (2000) Involvement of the Acute Phase Protein Alpha 1-Acid Glycoprotein in Nonspecific Resistance to a Lethal Gram-negative Infection. *J Biol Chem* 275: 14903–14909
- Hong F, Radaeva S, Pan HN, Tian Z, Veech R & Gao B (2004) Interleukin 6 alleviates hepatic steatosis and ischemia/reperfusion injury in mice with fatty liver disease. *Hepatology* 40: 933–941
- Hong YH, Varanasi US, Yang W & Leff T (2003) AMP-activated protein kinase regulates HNF4 α transcriptional activity by inhibiting dimer formation and decreasing protein stability. *Journal of Biological Chemistry* 278: 27495–27501
- Horvatits T, Drolz A, Rutter K, Roedl K, Langouche L, Van den Berghe G, Fauler G, Meyer B, Hülsmann M, Heinz G, *et al* (2017) Circulating bile acids predict outcome in critically ill patients. *Ann Intensive Care* 7

- Huck I, Gunewardena S, Espanol-Suner R, Willenbring H & Apte U (2019) Hepatocyte Nuclear Factor 4 Alpha Activation Is Essential for Termination of Liver Regeneration in Mice. *Hepatology* 70: 666–681
- Janz DR, Bastarache JA, Sills G, Wickersham N, May AK, Bernard GR & Ware LB (2013) Association between haptoglobin, hemopexin and mortality in adults with sepsis. *Crit Care* 17: R272
- Jin K, Ma Y, Manrique-Caballero CL, Li H, Emler DR, Li S, Baty CJ, Wen X, Kim-Campbell N, Frank A, *et al* (2020) Activation of AMP-activated protein kinase during sepsis/inflammation improves survival by preserving cellular metabolic fitness. *FASEB Journal* 34: 7036–7057
- Kikuchi S, Piraino G, O'Connor M, Wolfe V, Ridings K, Lahni P & Zingarelli B (2020) Hepatocyte-Specific Deletion of AMPK α 1 Results in Worse Outcomes in Mice Subjected to Sepsis in a Sex-Specific Manner. *Front Immunol* 11
- Kopczynski M, Rumieniczek I, Kulecka M, Statkiewicz M, Pysniak K, Sandowska-Markiewicz Z, Wojcik-Trechcinska U, Goryca K, Pyziak K, Majewska E, *et al* (2021) Selective extracellular signal-regulated kinase 1/2 (ERK1/2) inhibition by the SCH772984 compound attenuates in vitro and in vivo inflammatory responses and prolongs survival in murine sepsis models. *Int J Mol Sci* 22
- Kroy DC, Beraza N, Tschaharganeh DF, Sander LE, Erschfeld S, Giebeler A, Liedtke C, Wasmuth HE, Trautwein C & Streetz KL (2010) Lack of interleukin-6/glycoprotein 130/signal transducers and activators of transcription-3 signaling in hepatocytes predisposes to liver steatosis and injury in mice. *Hepatology* 51: 463–473
- Larsen R, Gozzelino R, Jeney V, Tokaji L, Bozza FA, Japiassú AM, Bonaparte D, Cavalcante MM, Chora Â, Ferreira A, *et al* (2010) A Central Role for Free Heme in the Pathogenesis of Severe Sepsis. *Sci Transl Med* 2: 51–71
- Libert C, Brouckaert P & Fiers W (1994) Protection by r Glycoprotein against Tumor Necrosis Factor-induced Lethality. *Journal of Experimental Medicine* 180: 1571–1575
- Libert C, Van Molle W, Brouckaert P & Fiers W (1996) alpha1-Antitrypsin inhibits the lethal response to TNF in mice. *J Immunol* 157: 5126–5129
- Lv C & Huang L (2020) Xenobiotic receptors in mediating the effect of sepsis on drug metabolism. *Acta Pharm Sin B* 10: 33–41 doi:10.1016/j.apsb.2019.12.003 [PREPRINT]
- Meuleman P, Libbrecht L, De Vos R, De Hemptinne B, Gevaert K, Vandekerckhove J, Roskams T & Leroux-Roels G (2005) Morphological and biochemical characterization of a human liver in a uPA-SCID mouse chimera. *Hepatology* 41: 847–856
- Van Molle W, Libert C, Fiers W & Brouckaert P (1997) Alpha 1-Acid Glycoprotein and Alpha 1-Antitrypsin Inhibit TNF-Induced but Not Anti-Fas-Induced Apoptosis of Hepatocytes in Mice. *J Immunol* 159: 3555–3565
- Moshage H (1997) Cytokines and the hepatic acute phase response. *Journal of Pathology* 181: 257–266
- Streetz KL, Luedde T, Manns MP & Trautwein C (2000) Interleukin 6 and liver regeneration. *Gut* 47: 309–312

- Sun K, Montana V, Chellappa K, Brelivet Y, Moras D, Maeda Y, Parpura V, Paschal BM & Sladek FM (2007) Phosphorylation of a conserved serine in the deoxyribonucleic acid binding domain of nuclear receptors alters intracellular localization. *Molecular Endocrinology* 21: 1297–1311
- Suzuki Y, Kami D, Taya T, Sano A, Ogata T, Matoba S & Gojo S (2023) ZLN005 improves the survival of polymicrobial sepsis by increasing the bacterial killing via inducing lysosomal acidification and biogenesis in phagocytes. *Front Immunol* 14
- Vega RB, Huss JM & Kelly DP (2000) The Coactivator PGC-1 Cooperates with Peroxisome Proliferator-Activated Receptor α in Transcriptional Control of Nuclear Genes Encoding Mitochondrial Fatty Acid Oxidation Enzymes. *Mol Cell Biol* 20: 1868–1876
- Vetö B, Bojcsuk D, Bacquet C, Kiss J, Sipeki S, Martin L, Buday L, Bálint BL & Arányi T (2017) The transcriptional activity of hepatocyte nuclear factor 4 alpha is inhibited via phosphorylation by ERK1/2. *PLoS One* 12
- Vida M, Gavito AL, Pavoń FJ, Bautista D, Serrano A, Suarez J, Arrabal S, Decara J, Romero-Cuevas M, De Fonseca FR, *et al* (2015) Chronic administration of recombinant IL-6 upregulates lipogenic enzyme expression and aggravates high-fat-diet-induced steatosis in IL-6-deficient mice. *DMM Disease Models and Mechanisms* 8: 721–731
- Viollet B, Kahn A & Raymondjean M (1997) Protein Kinase A-Dependent Phosphorylation Modulates DNA-Binding Activity of Hepatocyte Nuclear Factor 4. *Mol Cell Biol* 17: 4208–4219
- Wang YY, Ryg U, Dahle MK, Steffensen KR, Thiemermann C, Chaudry IH, Reinholt FP, Collins JL, Nebb HI, Aasen AO, *et al* (2011a) Liver X receptor protects against liver injury in sepsis caused by rodent cecal ligation and puncture. *Surg Infect (Larchmt)* 12: 283–289
- Wang YY, Ryg U, Dahle MK, Steffensen KR, Thiemermann C, Chaudry IH, Reinholt FP, Collins JL, Nebb HI, Aasen AO, *et al* (2011b) Liver X receptor protects against liver injury in sepsis caused by rodent cecal ligation and puncture. *Surg Infect (Larchmt)* 12: 283–289
- Van Wyngene L, Vanderhaeghen T, Timmermans S, Vandewalle J, Van Looveren K, Souffriau J, Wallaey C, Eggermont M, Ernst S, Van Hamme E, *et al* (2020) Hepatic PPAR α function and lipid metabolic pathways are dysregulated in polymicrobial sepsis. *EMBO Mol Med* 12
- Zhang W, Luo M, Zhou Y, Hu J, Li C, Liu K, Liu M, Zhu Y, Chen H & Zhang H (2021) Liver X receptor agonist GW3965 protects against sepsis by promoting myeloid derived suppressor cells apoptosis in mice. *Life Sci* 276

11th Jul 2024

Dear Prof. Libert,

Thank you for the submission of your revised manuscript to EMBO Molecular Medicine. I am pleased to inform you that we will be able to accept your manuscript pending the following final amendments:

- 1) Authors: Please provide current email address for Bart Roman.
 - 2) In the main manuscript file, please do the following:
 - Please address all comments suggested by our data editors listed below:
 - o Figure legends:
 1. Please define the annotated p values ****/*** as well as provide the exact p-values for the same in the legend of figure 4c; EV 1b-c; as appropriate.
 2. Please note that the exact p values are not provided in the legends of figures 3f; 5b-d, f, h-k; 6b, d, g, i-o; 7b-c, e-n; EV 4a, d-e, g.
 3. Please indicate the statistical test used for data analysis in the legends of figures 1c-e; 2i-j; 3d; 4a, e-g; 6f; EV 2b; EV 3; EV 5b-c.
 4. Please note that in figures 2a, e; there is a mismatch between the annotated p values in the figure legend and the annotated p values in the figure file that should be corrected.
 5. Please note that the box plot needs to be defined in terms of minima, maxima, centre, bounds of box and whiskers, and percentile in the legend of figure 3f.
 6. Please note that information related to n is missing in the legends of figures 3f; 4h; 6b; EV 1b-c.
 7. Please note that the error bars are not defined in the legends of figures 4h; EV 1b-c.
 - Remove all supplementary information.
 - Rename "Competing interests" to "Disclosure and competing interests statement". We updated our journal's competing interests policy in January 2022 and request authors to consider both actual and perceived competing interests. Please review the policy <https://www.embopress.org/competing-interests> and update your competing interests if necessary. Also, please add the following statement: "Claude Libert is a member of the Advisory Editorial Board of EMBO Molecular Medicine. This has no bearing on the editorial consideration of this article for publication.
 - Author contributions: Please remove it from the manuscript and specify author contributions in our submission system. CRediT has replaced the traditional author contributions section because it offers a systematic machine-readable author contributions format that allows for more effective research assessment. You are encouraged to use the free text boxes beneath each contributing author's name to add specific details on the author's contribution. More information is available in our guide to authors:
<https://www.embopress.org/page/journal/17574684/authorguide#authorshipguidelines>
 - Please include structured Methods section that includes a Reagents and Tools Table followed by a Methods and Protocols section. More information on how to adhere to this format as well as downloadable templates (.docx) for the Reagents and Tools Table can be found in our author guidelines: <https://www.embopress.org/page/journal/17574684/authorguide#structuredmethods>
An example of a paper with Structured Methods can be found here:
<https://www.embopress.org/doi/full/10.1038/s44320-024-00037-6#sec-4>
 - Indicate in legends number and nature of replicates and exact p= values, not a range, along with the statistical test used. To keep the figures "clear" some authors found providing an Appendix table Sx with all exact p-values preferable. You are welcome to do this if you want to.
 - In Methods, add the following paragraph:
- Graphics:
(some of the... OR Figure #... OR synopsis) Graphics were created with BioRender.com.
- Please remove the reference to BioRender from Acknowledgments.
- In data availability statement please add the specific URL for S-BIAD1239 dataset.
 - 3) Appendix: Please correct the nomenclature in the legends to "Appendix Fige S1" etc. and "Appendix Table S1" etc.
 - 4) The Paper Explained: Please add it to the main manuscript file with following headings "Problem", "Results" and "Impact". Also, please shorten the "Results" part by removing the first paragraph which seems to describe the problem. This paragraph could replace the "Medical issue" text, which in the current form reads as a general introduction and does not clearly describe the problem you are addressing. In the "Clinical impact" remove repetition of results and only highlight the clinical impact of the study. Please refer to any of our published primary research articles for an example.
 - 5) Synopsis:
 - Synopsis image: Please resize the image to 550 px-wide x (250-400)-px high and upload it as a high-resolution jpeg file. Also, please increase font size in the image.
 - Please check your synopsis text and image before submission with your revised manuscript. Please be aware that in the proof stage minor corrections only are allowed (e.g., typos).
 - 6) As part of the EMBO Publications transparent editorial process initiative (see our Editorial at

<http://embomolmed.embopress.org/content/2/9/329>), EMBO Molecular Medicine will publish online a Review Process File (RPF) to accompany accepted manuscripts. This file will be published in conjunction with your paper and will include the anonymous referee reports, your point-by-point response and all pertinent correspondence relating to the manuscript. Let us know whether you agree with the publication of the RPF and as here, if you want to remove or not any figures from it prior to publication. Please note that the Authors checklist will be published at the end of the RPF.

7) Please provide a point-by-point letter INCLUDING my comments as well as the reviewer's reports and your detailed responses (as Word file).

I look forward to reading a new revised version of your manuscript as soon as possible.

Yours sincerely,

Zeljko Durdevic

*** Instructions to submit your revised manuscript ***

1) a .docx formatted version of the manuscript text (including Figure legends and tables)

2) Separate figure files*

3) supplemental information as Expanded View and/or Appendix. Please carefully check the authors guidelines for formatting Expanded view and Appendix figures and tables at <https://www.embopress.org/page/journal/17574684/authorguide#expandedview>

4) a letter INCLUDING the reviewer's reports and your detailed responses to their comments (as Word file).

5) The paper explained: EMBO Molecular Medicine articles are accompanied by a summary of the articles to emphasize the major findings in the paper and their medical implications for the non-specialist reader. Please provide a draft summary of your article highlighting

6) For more information: There is space at the end of each article to list relevant web links for further consultation by our readers. Could you identify some relevant ones and provide such information as well? Some examples are patient associations, relevant databases, OMIM/proteins/genes links, author's websites, etc...

7) Author contributions: the contribution of every author must be detailed in a separate section.

8) EMBO Molecular Medicine now requires a complete author checklist (<https://www.embopress.org/page/journal/17574684/authorguide>) to be submitted with all revised manuscripts. Please use the checklist as guideline for the sort of information we need WITHIN the manuscript. The checklist should only be filled with page numbers where the information can be found. This is particularly important for animal reporting, antibody dilutions (missing) and exact values and n that should be indicated instead of a range.

9) Every published paper now includes a 'Synopsis' to further enhance discoverability. Synopses are displayed on the journal webpage and are freely accessible to all readers. They include a short stand first (maximum of 300 characters, including space) as well as 2-5 one sentence bullet points that summarise the paper. Please write the bullet points to summarise the key NEW findings. They should be designed to be complementary to the abstract - i.e. not repeat the same text. We encourage inclusion of key acronyms and quantitative information (maximum of 30 words / bullet point). Please use the passive voice. Please attach these in a separate file or send them by email, we will incorporate them accordingly.

You are also welcome to suggest a striking image or visual abstract to illustrate your article. If you do please provide a jpeg file 550 px-wide x 300-600px high.

10) A Conflict of Interest statement should be provided in the main text

11) Please note that we now mandate that all corresponding authors list an ORCID digital identifier. This takes <90 seconds to complete. We encourage all authors to supply an ORCID identifier, which will be linked to their name for unambiguous name identification.

Currently, our records indicate that the ORCID for your account is 0000-0001-6408-036X.

Link Not Available

12) Include a Reagents and Tools Table as part of the Methods section, which can be downloaded from our author guidelines (<https://www.embopress.org/page/journal/17574684/authorguide#structuredmethods>)

Photos 400-800 DPI

*Additional important information regarding figures and illustrations can be found at <https://bit.ly/EMBOPressFigurePreparationGuideline>. See also figure legend preparation guidelines: <https://www.embopress.org/page/journal/17574684/authorguide#figureformat>

***** Reviewer's comments *****

Referee #2 (Remarks for Author):

My concerns were addressed properly in the revised version. The paper provides strong evidence for the transcriptional hierarchy involved in the regulation of sepsis-induced genes. The requirement for HNF4 regulatory function is clearly demonstrated.

Referee #3 (Remarks for Author):

The revised version has properly addressed all concerns.

The authors addressed the minor editorial issues.

13th Aug 2024

Dear Prof. Libert,

We are pleased to inform you that your manuscript is accepted for publication and is now being sent to our publisher to be included in the next available issue of EMBO Molecular Medicine.
